# Transient remodeling of gut metabolism supports juvenile growth and adult fitness in *Drosophila*

Clara Lefranc [1,2], Arnaud Fichant[3] & Gilles Storelli [1,2,3] ✉

Animals develop through discrete life stages with specific goals, such as growth or reproduction. Achieving these goals may require temporary changes in the function of certain organs, but these adaptations remain poorly understood. In *Drosophila*, the juvenile phase is characterized by a several-hundred-fold increase in body mass, which culminates in a rapid growth spurt before puberty. Here, we show that this growth spurt is supported by acute remodeling of gut metabolism. Midway through the juvenile phase, steroid hormone and hepatocyte nuclear factor 4 induce digestive function, lipid metabolism and nutrient export in the intestine, thereby accelerating growth and maturation. Gut metabolic remodeling ceases at puberty but has a lasting impact on physiology, enhancing adult reproductive fitness and resilience to environmental stress. Our work identifies an endocrine-metabolic axis that synchronizes gut function with developmental demands, and provides insights into how systemic signals dynamically remodel organ metabolism to optimize life history strategies.

The life cycle of most animals includes phases dedicated to growth, maturation and reproduction. The endocrine pathways that control the transitions between these phases are well characterized. In contrast, far less is known about how metabolism and physiology are regulated between these developmental transitions. For example, juvenile organisms have nutritional, metabolic and physiological needs that differ from those of non-growing adults. These needs may even fluctuate during infancy, as in humans, where growth occurs in discrete phases[1]. Such changing needs may require transient, organ-specific adaptations to be met. In parallel, perturbations in juvenile nutrition and metabolism not only influence growth, but also the onset of puberty and adult health[2]. Thus, a systematic understanding of organ metabolism throughout the life cycle could reveal fundamental principles of developmental biology, but also periods of nutritional and metabolic vulnerability that can leave lifelong impacts on physiology[3,4].

*Drosophila* are an ideal model to explore these questions. Their organs have functions analogous to those of more complex animals[5,6].

Their juvenile phase, metamorphosis (the equivalent of puberty) and adulthood are delimited by clear developmental landmarks. The juvenile phase is divided into three larval stages (L1, L2 and L3), each of which are separated by a molt. These molts and metamorphosis are triggered by pulses of the steroid hormone ecdysone[7]. This hormone acts through the ecdysone receptor (EcR), a member of the nuclear receptor (NR) superfamily of ligand-regulated transcription factors, to coordinate tissue-specific developmental programs[7]. In parallel, other NRs, including hepatocyte nuclear factor 4 (HNF4), coordinate metabolic shifts that support developmental progression[8–10]. HNF4 binds to fatty acids and plays an evolutionary conserved role in the transcriptional regulation of lipid metabolism[10–13]. This includes pathways involved in the catabolism of fat stores, inter-organ lipid trafficking, and the synthesis of specific lipid classes and their derivatives[10–14]. In *Drosophila*, HNF4 induces a transcriptional program in subepidermal cells called oenocytes once metamorphosis is complete[10]. This prompts them to convert circulating lipids into hydrocarbons, which act as a waterproof coating for the cuticle[10]. This process is essential

[1]University of Cologne, Faculty of Mathematics and Natural Sciences, Cologne Excellence Cluster on Aging and Aging-Associated Diseases (CECAD), Cologne, Germany. [2]University of Cologne, Faculty of Mathematics and Natural Sciences, Institute for Genetics, Cologne, Germany. [3]Heidelberg University, Centre for Organismal Studies (COS) Heidelberg, Heidelberg, Germany. ✉e-mail: gilles.storelli@cos.uni-heidelberg.de

for desiccation resistance and survival in newly eclosed adults, which leave the protection of their pupal case to adopt a nomadic lifestyle[10]. In gravid female mosquitoes, ecdysone can induce the mobilization of fat stores by promoting *HNF4* expression[15]. This epistatic relationship in mature insects suggests that EcR and HNF4 may cooperate more extensively during development to regulate lipid metabolism in specific organs. These regulations may be particularly important at the onset of adulthood[10], but also during periods of high anabolic demand, such as the larval stages of *Drosophila*, when body mass increases 200-fold within a few days[16].

Here, we show that the final larval instar in *Drosophila* hosts an intense growth spurt supported by metabolic remodeling of the gut. Upon entry into the last larval instar, EcR and HNF4 trigger acute metabolic changes in the intestine, with sharp increases in digestion, lipid metabolism and nutrient transport. This metabolic switch fulfils the metabolic demands of rapid growth and ensures timely metamorphosis. Gut metabolic remodeling ceases at the onset of metamorphosis, but has a lasting impact on physiology. It promotes

reproductive fitness and stress resilience in adults, demonstrating that early-life metabolic transitions can set long-term physiological trajectories. From a broader perspective, this study contributes to our global understanding of how hormonal signals regulate metabolism in specific organs to fulfill developmental demands and optimize life history strategies.

## Results

### Growth accelerates midway through larval life

The *Drosophila* juvenile phase is divided into three larval stages, which together last approximately five days, after which metamorphosis begins (Fig. 1a, b). We scored larval growth by measuring body volume during larval life[17]. Consistent with previous studies, we observed a 73-fold increase in body volume during this period (Fig. 1c)[17–19]. However, growth is not uniform: it accelerates dramatically after the third day, coinciding with the transition from the second to the third larval instar (Fig. 1a, c). Protein, a metabolite commonly used as an indicator of biomass, follows a similar trend: it increases sharply after the third day

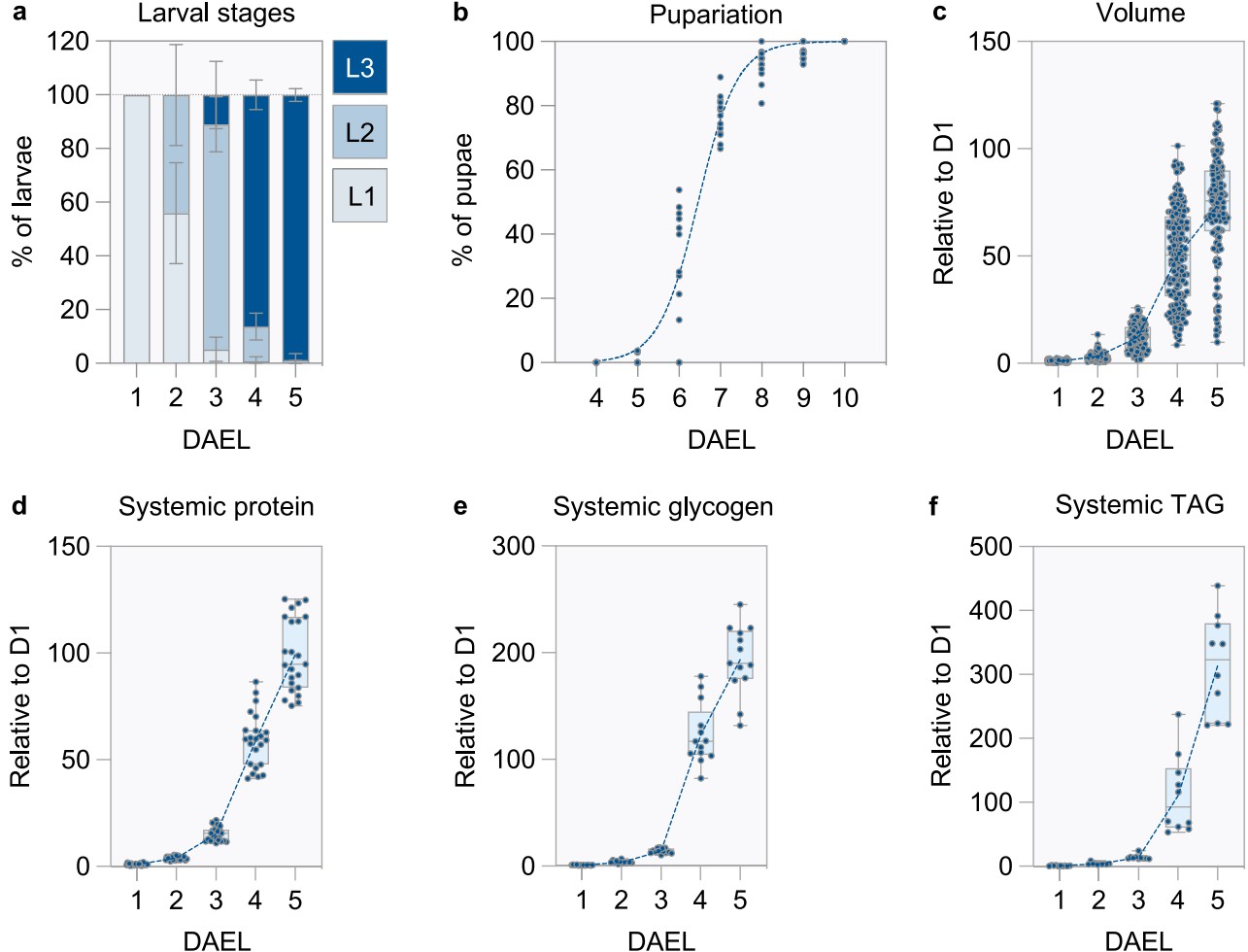

**Fig. 1 | Growth accelerates midway through larval life. a–f** Growth and maturation was assessed in $w^{1118}$ larvae. DAEL: days after egg laying. **a** The relative proportion of larval stages was scored between D1-5AEL. Bar graphs represent the mean and standard deviation. $n = 11$ (D1-D4AEL) and 10 (D5AEL) biological replicates from two independent experiments (one vial seeded with 40 embryos is considered one biological replicate). The gray dotted line represents 100% of the total. **b** Pupariation was scored between D4-10AEL, and is presented as a percentage of the total number of pupae. The sigmoid curve represents the non-linear least square regression fit of the data. Median time to pupariation, extrapolated from the non-linear fit analysis = 6.46 days. $n = 13$ biological replicates from three independent experiments (one vial seeded with 40 embryos is considered one biological replicate). **(c)** Larval

volume between D1-D5AEL. $n = 186, 198, 152, 170$ and 152 larvae for D1, D2, D3, D4 and D5AEL, respectively. One representative experiment out of two is shown. Total (**d**) protein, **e** glycogen and **f** triglyceride (TAG) content was scored between D1-D5AEL. **d** $n = 23$, **e** 13 and **f** 10 biological replicates, each containing several animals, from **d** five, **e** three and **f** two independent experiments, respectively. See the methods for details about the contents of the biological replicates used in these assays. **b–f** Dots represent individual biological replicates or specimens. **c–f** Boxplots extend from the 25th to 75th percentile, whiskers extend from the minimum to maximum values, and the median is depicted as a line. Values are plotted relative to those of one-day-old larvae. A blue dashed line connects mean values for each day. Source data are provided as a Source Data file.

of development, with an overall increase of 99-fold (Fig. 1d). Glycogen and triglyceride show similar temporal evolutions, with 193- and 313-fold overall increases, respectively (Fig. 1e, f). Thus, *Drosophila* experience a growth spurt during the last larval instar.

### The intestinal transcriptome is remodeled in L3 larvae

Growth is closely linked to nutrition[20], and *Drosophila* feed continuously during larval development (Supplementary Fig. 1a)[21]. Therefore, we hypothesized that physiological and metabolic shifts in the gut, which plays a fundamental role in nutrient digestion and assimilation, could support increased anabolism in L3 larvae[22]. We performed bulk mRNA-sequencing in the midgut (the insect equivalent of the vertebrate small intestine) at timepoints surrounding growth acceleration. For this, we isolated midguts from two-, three- and four-day old larvae, which corresponds to early L2, late L2 and early L3 stage, respectively (Fig. 1a). We compiled a list of genes that are differentially expressed ($|\log2FC| \geq 0.5$ and adjusted p-value < 0.05) in at least one of the following comparisons: late L2 versus early L2 midguts, and early L3 versus late L2 midguts (Supplementary Data 1). 429 transcripts in this list show significant changes in their levels during the L2 stage (late L2 versus early L2 midguts, $|\log2FC| > 0$ and adjusted p-value < 0.05, [Supplementary Data 1]). In contrast, 1469 transcripts show significantly altered levels between late L2 and early L3 midguts, suggesting a profound remodeling of intestinal function upon entering the third larval instar (Supplementary Data 1). Hierarchical clustering of the 1000 most variable transcripts across the three timepoints confirmed this pattern: the levels of most transcripts is stable during the L2 stage, but undergo sharp up- or down-regulations after the entry into the L3 stage (clusters A-B and D-E, respectively, [Fig. 2a and Supplementary Data 1]). We performed gene ontology (GO) term enrichment analysis to determine if specific biological processes are overrepresented in these clusters. Clusters A and B, which regroup transcripts upregulated in L3 midguts, show an enrichment for terms related to lipid metabolism (Fig. 2a, b and Supplementary Data 2). Cluster C, which regroups transcripts mildly downregulated after the early L2 stage, does not show any significant enrichment (Fig. 2a and Supplementary Data 2). Finally, Clusters D and E, which regroup transcripts downregulated in L3 midguts, show an enrichment for terms related to ribosome biogenesis and cuticle formation (Fig. 2a, b and Supplementary Data 2). Thus, systemic growth acceleration is accompanied with a transcriptional shift in the midgut, with likely alterations in ribosome function, cuticle development and lipid metabolism.

### Lipid digestion, synthesis and export are induced in L3 guts

For simplicity, "late L2" and "early L3" will be referred to as "L2" and "L3", respectively, throughout the rest of the text. Our transcriptomic analysis revealed a coordinated increase in transcripts related to lipid metabolism in the L3 midgut (Fig. 2b and Supplementary Data 2). We validated this signature using RT-qPCR and made similar observations in another control background (Supplementary Fig. 2a). Intestinal lipid processing involves several steps[22]. First, digestive lipases breakdown dietary lipids into fatty acids that can be absorbed by enterocytes (ECs)(Fig. 3a)[22]. We have compiled a list of putative digestive enzymes in *Drosophila* (Supplementary Data 3, see methods for details). Most genes encoding digestive lipases are transcriptionally induced in the L3 midgut, indicating an activation of lipid digestion (Fig. 3b, Supplementary Fig. 2a and Supplementary Data 3). Once absorbed by ECs, free fatty acids (FFA) can follow three major fates: they can be oxidized for energy production, exported to peripheral tissues in lipoproteins, or re-esterified for storage in lipid droplets (Fig. 3a)[22]. The latter can be mobilized by intracellular lipases to re-generate FFA[14,22]. Numerous genes encoding proteins involved in lipid synthesis, breakdown and transport are transcriptionally induced in the L3 midgut, and we made similar

observations in another genetic background (Fig. 3c, Supplementary Fig. 2a and Supplementary Data 1-2). We sought to determine the functional impact of these transcriptional changes on midgut lipid metabolism. Consistent with the increased expression of digestive and intracellular lipases, we observed a 2.8-fold increase in triglyceride lipase activity between L2 and L3 midguts after normalization to tissue size (Fig. 3d). These changes are accompanied by a 3.4-fold increase in free glycerol levels, a metabolite generated during intracellular triglyceride catabolism (Fig. 3e). Therefore, increased digestion of dietary fat occurs alongside enhanced lipid droplet mobilization, leading to a slight increase in intracellular lipid levels in midguts between the L2 and L3 stage (Fig. 3f, g). Simultaneous increases in the uptake of dietary fats and in the mobilization of lipid droplets could favor FFA incorporation into lipoproteins and export to peripheral tissues (Fig. 3a)[14]. Consistent with this model, circulating triglyceride levels and systemic fat stores increase after animals enter into the L3 stage (Figs. 3h and 1f). Carbohydrate digestion is also induced in the L3 midgut, potentially providing additional resources for intestinal lipid synthesis and systemic fat storage (Supplementary Fig. 2c, d and Supplementary Data 3). Taken together, these observations support a model in which intestinal lipid metabolism is remodeled midway through larval life to speed up the accumulation of fat stores.

### Ecdysone triggers a switch in intestinal lipid metabolism

Next, we sought to identify the transcription factors (TFs) responsible for this acute change in intestinal metabolism. To this end, we performed TF motif enrichment analysis in the vicinity of genes belonging to the clusters identified in (Fig. 2a). This analysis on the clusters regrouping upregulated transcripts resulted in a list of candidate TFs, including EcR and its downstream targets br, E74, E75 and E78 (Supplementary Data 4). Given that ecdysone signaling dictates the transition from the L2 to L3 stage, this pathway may participate in the developmental induction of lipid metabolism in the gut. Consistent with this hypothesis, increased expression of *EcR, br, Eo* and *shd* in L3 midguts demonstrates temporal induction of ecdysone signaling, and EcR is expressed in regions of the midgut that accumulate lipids (Fig. 4a–c and Supplementary Fig. 3a). To test a role for EcR in the developmental induction of intestinal lipid metabolism, we expressed a dominant-negative form of this nuclear receptor (EcR[DN]) in ECs[23]. As expected, this manipulation blocks the induction of ecdysone signaling in L3 midguts (Fig. 4d). It also has global, negative impacts on the induction of transcripts involved in lipid metabolism in L3 midguts (Fig. 4d). Importantly, these effects are not due to decreased midgut growth or EC endoreplication (Supplementary Fig. 3b–f). Since EcR[DN] blocks the developmental induction of lipid-related transcripts, we checked its impacts on intestinal lipid levels. Under *EcR[DN]* expression, the midgut accumulates lipids in L3 larvae (Fig. 4e, f). This observation could be explained by a defect in lipid export, as inhibition of lipoprotein synthesis has a similar effect (Supplementary Fig. 3g, h). Consistent with this hypothesis, suppression of ecdysone signaling in ECs depletes circulating lipid levels, and reduces fat accumulation after entering the L3 stage (Fig. 4g, h). Several of these effects are recapitulated when *EcR* is silenced with RNAi in ECs (Supplementary Fig. 4a, b). Therefore, ecdysone triggers a developmental switch in gut metabolism to maximize lipid export and the buildup of systemic fat stores.

### HNF4 is necessary for intestinal metabolic remodeling

EcR may act in parallel to, or via, other TFs to exert its effects on lipid metabolism. In particular, binding motifs for HNF4 are present in the vicinity of genes whose expression is induced in L3 midguts (Supplementary Data 4). HNF4 is also induced during development and detected in regions of the larval midgut that accumulate lipids, further supporting a role in the developmental induction of lipid metabolism

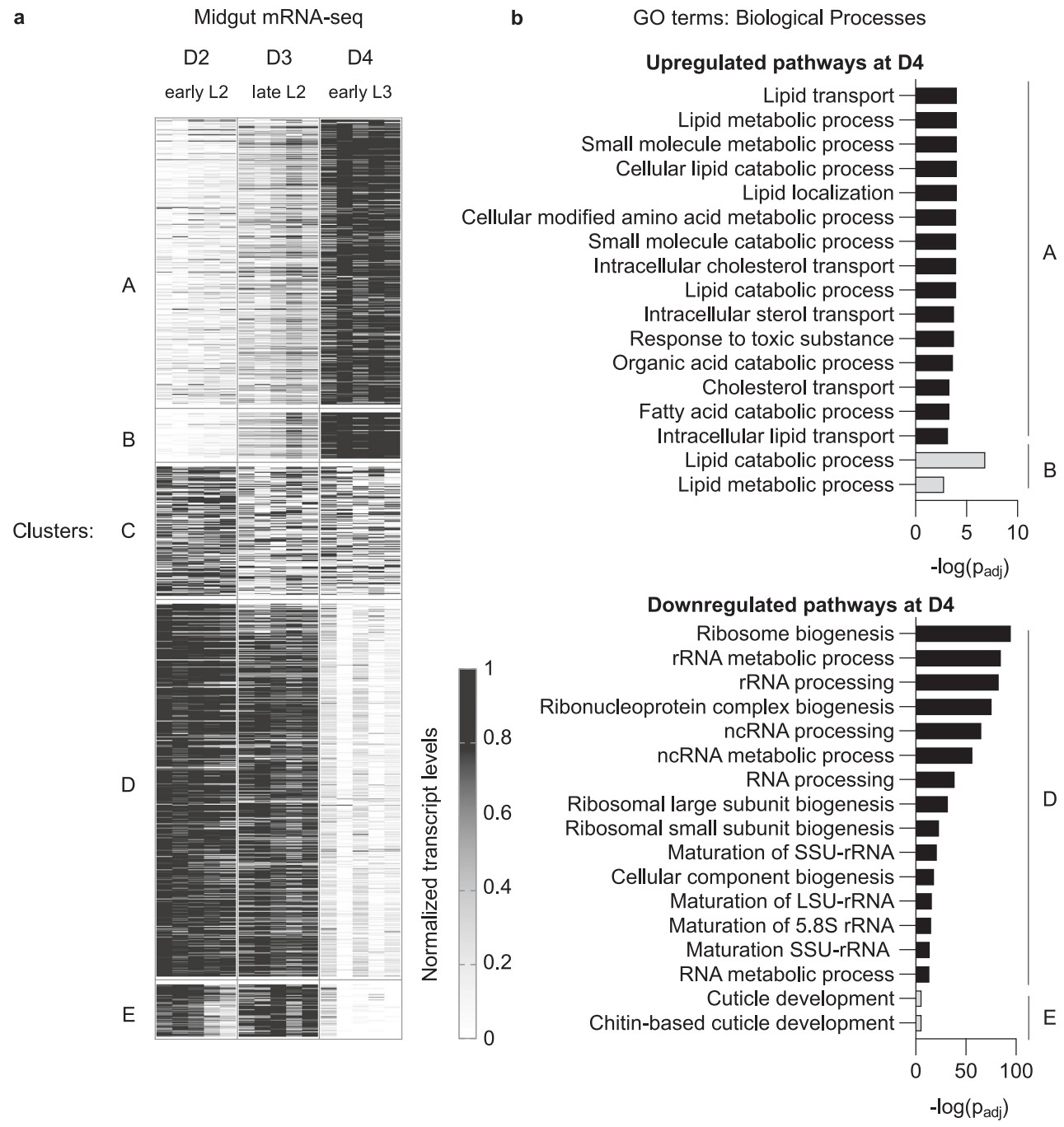

**Fig. 2 | The intestinal transcriptome is remodeled in L3 larvae. a, b** mRNA-seq was performed on midguts isolated from $w^{1118}$ larvae at D2, D3 and D4AEL (corresponding to early L2, late L2 and early L3, respectively). **a** The heatmap shows the 1000 most variable transcripts across all samples, after hierarchical clustering and min-max normalization. Transcript levels are represented using a greyscale (minimal and maximal values are represented by white and black, respectively, see methods for details). $n = 5$ biological replicates. **b** Gene Ontology (GO) term enrichment analysis was performed on the clusters of transcripts identified using hierarchical clustering. The panels depict the biological processes that are significantly enriched in the different clusters, following a one-sided Fisher's exact test and Benjamini-Hochberg procedure to account for multiple testing. Upper panel: clusters A-B, lower panel: clusters D-E.

(Fig. 5a–c and Supplementary Fig. 5a). To test this possibility, we silenced *Hnf4* in ECs using RNAi, and performed bulk mRNA-sequencing in L2 and L3 midguts (Supplementary Data 5). As previously observed, numerous genes involved in lipid metabolism are transcriptionally induced upon entry into the L3 stage in control midguts (Supplementary Data 5). 45% of these transcripts depend on HNF4 for their induction (Cluster A in [Fig. 5d and Supplementary Data 5]). We validated several of these regulatory relationships with RT-qPCR (Supplementary Fig. 5b). In parallel, *Hnf4* overexpression in otherwise wild-type ECs further increases the expression of several of these lipid-related genes in L3

midguts (Supplementary Fig. 5c). Therefore, HNF4 drives, at least in parts, intestinal metabolic remodeling during larval development. Among its metabolic roles, HNF4 promotes intestinal lipid export at the expense of storage in lipid droplets[14]. Consistent with this function, suppression of *Hnf4* leads to fat retention in the midgut and depletion of circulating lipids in the hemolymph of L3 larvae (Fig. 5e–g). As a result, these animals show reduced fat accumulation after entering the L3 stage (Fig. 5h). Altogether, these analyses show that HNF4 participates in intestinal metabolic remodeling and supports fat storage during the last larval instar.

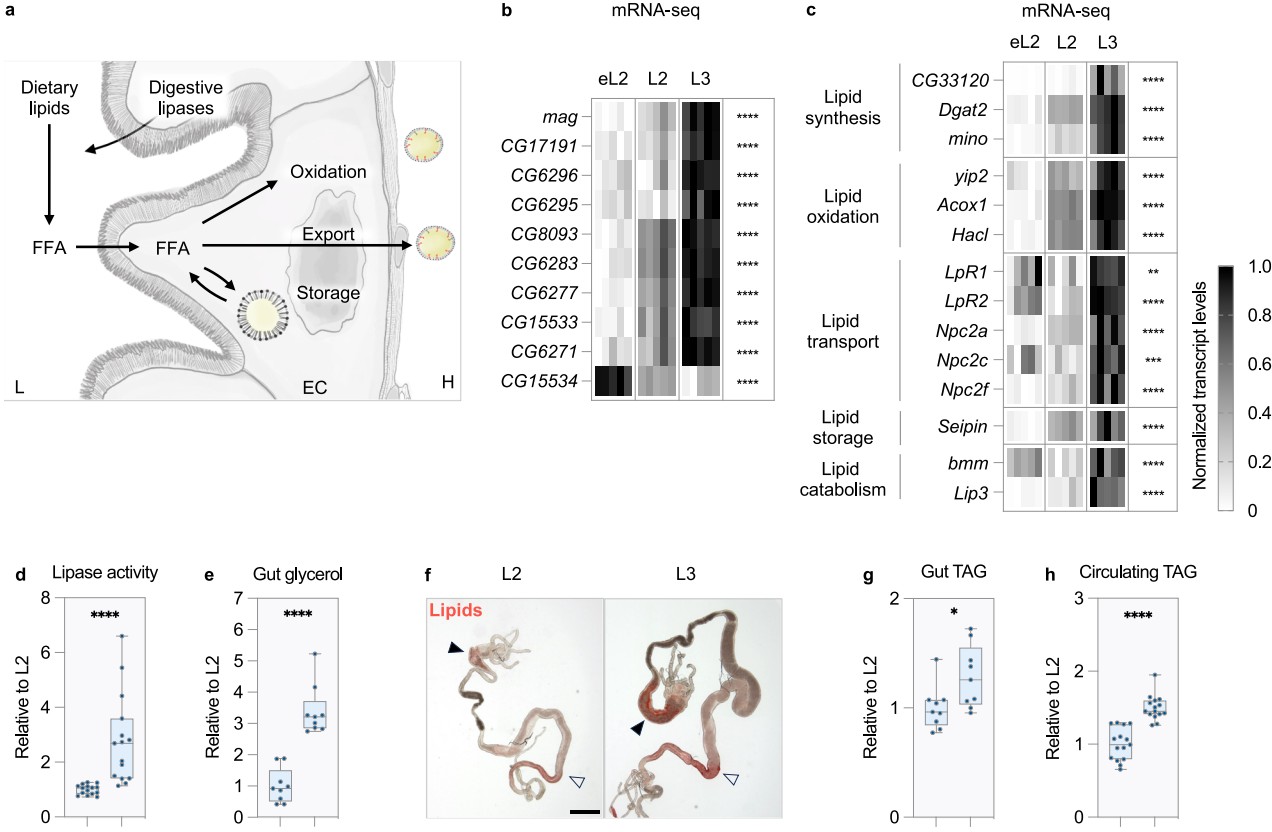

**Fig. 3 | Lipid digestion, synthesis and export are induced in L3 midguts.**
**a** Representation of intestinal lipid handling. EC: enterocytes, FFA: free fatty acids, H: hemolymph, L: lumen. Illustration adapted from this article was published in Cell Reports, Volume 43, Maximilian C. Vonolfen, Fenja L. Meyer zu Altenschildesche, Hyuck-Jin Nam, Susanne Brodesser, Akos Gyenis, Jan Buellesbach, Geanette Lam, Carl S. Thummel, Gilles Storelli, Drosophila HNF4 acts in distinct tissues to direct a switch between lipid storage and export in the gut, Copyright Elsevier (2024) under a CC BY license: https://creativecommons.org/licenses/by/4.0/. (**b-h**) Experiments with $w^{1118}$ larvae. Levels of transcripts involved in (**b**) lipid digestion and (**c**) other aspects of lipid metabolism in early L2 (eL2), late L2 (L2) and early L3 (L3) midguts. Transcript levels are represented using a greyscale after min-max normalization (minimal and maximal values represented by white and black, respectively, see methods for details). $n = 5$ biological replicates per timepoint. **d** Triglyceride lipase activity, (**e**) free glycerol and (**g**) triglyceride content in L2 and L3 midguts. Values normalized to protein content and relative to L2 midguts. $n =$ (**d**) 15, (**e**) 9, and (**g**) 9

biological replicates per genotype, from three independent experiments. Biological replicates contain several organs, see the methods for details. (**f**) Lipid stains in L2 and L3 midguts with Oil Red O. Closed arrowhead: anterior midgut, open arrowhead: posterior midgut. Representative image from $n = 30$ midguts per timepoint from three independent experiments. Scale bar = 500 μm. **h** Circulating triglyceride levels, normalized to hemolymph volume and relative to L2 larvae. $n = 15$ biological replicates per genotype from three independent experiments. Hemolymph was collected from several animals to make up one biological replicate. See the methods for details. **d, e, g, h** Dots represent biological replicates. Boxplots extend from the 25th to 75th percentile, whiskers extend from minimum to maximum, median is depicted as a line. **b–e, g, h** Asterisks indicate statistically significant difference between timepoints ((**b, c**) 1-way ANOVA, (**d, e**) Two-sided Mann-Whitney test or (**g, h**) two-sided Student's $t$ test). **d, e** $p \leq 0.0001$, (**g**) $p = 0.026$. *$p \leq 0.05$, **$p \leq 0.01$, ***$p \leq 0.001$, ****$p \leq 0.0001$. Source data are provided as a Source Data file.

## EcR acts via several pathways to support fat storage in L3

Suppression of EcR and HNF4 in ECs have similar effects on intestinal and systemic lipid levels, suggesting genetic interaction between these two NRs (Figs. 4 and 5). Analysis of publicly available ChIP-seq data via the TF2TG portal indicates that EcR and its dimerization partner USP bind several times in the vicinity of the *Hnf4* locus in *Drosophila* (Supplementary Data 6)[24]. Eip74EF, Eip75B, Eip93F, Blimp-1 and crol, which are well-established transcriptional targets of EcR, also bind in the vicinity of the *Hnf4* locus (Supplementary Data 6)[24]. Therefore, EcR may regulate *Hnf4* expression in direct and indirect ways. Consistent with this hypothesis, *Hnf4* expression increases in the midgut upon entry into the L3 stage, and this induction is blocked by the inhibition of ecdysone signaling (Fig. 5i). In contrast, silencing or overexpressing *Hnf4* does not affect ecdysone signaling in L3 midguts (Supplementary Fig. 5d, e). Therefore, EcR promotes *Hnf4* expression in the L3 midgut, and HNF4 could in turn induce several pathways of lipid metabolism downstream of EcR. However, *Hnf4* overexpression does not rescue systemic fat levels in L3 larvae

with inhibition of ecdysone signaling in ECs (Fig. 5j). Therefore, HNF4 is necessary in the gut for rapid fat storage in L3 larvae, but is not sufficient on its own to support this process when ecdysone signaling is inhibited in ECs. Taken together, these data support a model in which EcR acts through multiple pathways, including HNF4 signaling, to drive intestinal metabolic remodeling and rapid fat storage in L3 larvae.

## EcR and HNF4 act in ECs to accelerate growth and maturation

As lipids are essential building blocks, the developmental induction of intestinal lipid metabolism could also support systemic growth. Consistent with this hypothesis, inhibiting EcR or HNF4 in ECs reduces body growth specifically after animals enter the L3 stage (Fig. 6a, b and Supplementary Fig. 6a). These effects could be explained by the leanness of these animals (Figs. 4h, 5h and Supplementary Fig. 4b). However, animals with inhibition of EcR or HNF4 in ECs accumulate less protein than controls after entering the L3 stage, demonstrating a general defect in body growth (Fig. 6c, d and Supplementary Fig. 6b).

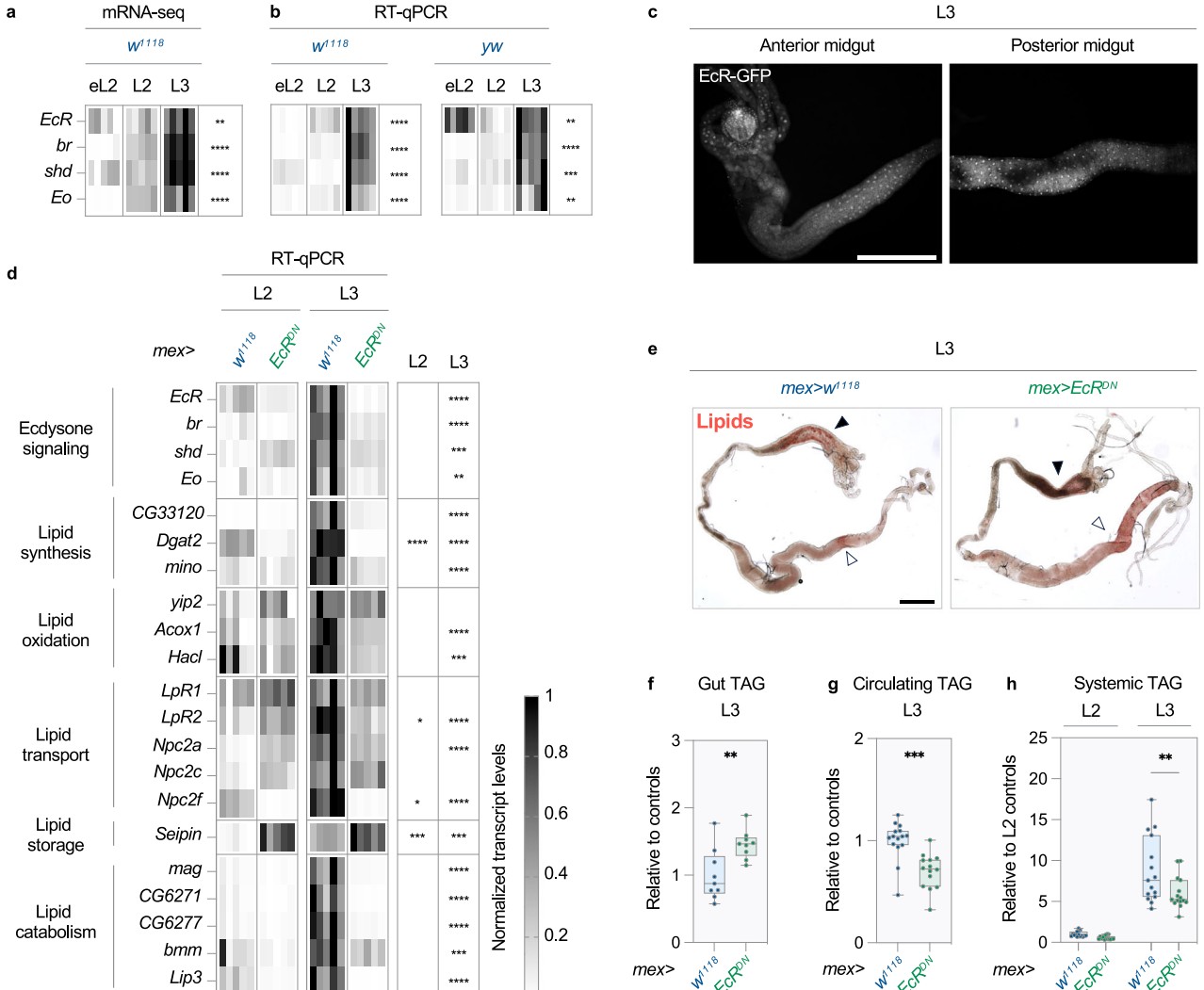

**Fig. 4 | Ecdysone triggers a switch in intestinal lipid metabolism. a–h** Early L2 (eL2), late L2 (L2) or early L3 (L3) larvae were analyzed. **a, b** Levels of transcripts involved in ecdysone signaling in $w^{1118}$ and $yw$ midguts. $n = 5$ biological replicates. **b** One experiment out of three is shown. **c** Anti-GFP antibodies were used to visualize EcR in *EcR-GFP* midguts. Representative image from two independent experiments ($n = 20$ midguts). Scale bar = 500 μm. An entire midgut is shown in (Supplementary Fig. 3a). **d–h** EcR$^{DN}$ was expressed in ECs with the *mex-GAL4* driver (*mex > EcR$^{DN}$*). *mex > $w^{1118}$* animals were used as controls. **d** Levels of transcripts involved in ecdysone signaling and lipid metabolism in midguts. $n = 5$ biological replicates. One experiment out of three is shown. **e** Oil Red O stains of neutral lipids. The anterior and posterior midgut are indicated with closed and open arrowheads, respectively. Representative image from three independent experiments ($n = 30$ midguts). Scale bar = 500 μm. Triglyceride content in **f** midguts (normalized to protein) and (**g**) hemolymph (normalized to volume). Values are presented relative to controls. **f** $n = 9$ (both genotypes) and **g** $n = 15$ (*mex > $w^{1118}$*) and 14 (*mex > EcR$^{DN}$*) biological

replicates from three independent experiments. **h** Total triglyceride content in larvae. Values presented relative to L2 controls. $n = 15$ biological replicates per genotype and timepoint from three independent experiments. **a, b, d, f–h** Biological replicates generated from several animals/organs, see methods. **a, b, d** Heatmaps depict transcript levels and statistically significant differences between conditions. Transcript levels are represented using a greyscale after min-max normalization (see methods for details). **f–h** Dots represent biological replicates, boxplots extend from the 25th to 75th percentile, whiskers from minimum to maximum, median depicted as a line. **a, b, d, f–h** Asterisks indicate statistically significant differences (**a, b**) across timepoints ((**a**) 1-way ANOVA, **b** Kruskal-Wallis test), **d, h** between genotypes at a given stage (2-way ANOVA with Šidák's multiple comparisons) or (**f, g**) between genotypes ((**f**) two-sided Student's $t$ test and **g** two-sided Mann-Whitney test). **f** $p = 0.0065$, **g** $p = 0.0001$, **h** $p = 0.8592$ (L2) and $p = 0.0029$ (L3). *$p \le 0.05$, **$p \le 0.01$, ***$p \le 0.001$, ****$p \le 0.0001$. Source data are provided as a Source Data file.

*Hnf4* overexpression with simultaneous inhibition of ecdysone signaling does not rescue total protein levels in L3 larvae (Supplementary Fig. 6c). Therefore, and as previously observed for fat stores, HNF4 is necessary in ECs for faster growth, but is not sufficient to support this process when ecdysone signaling is inhibited. In *Drosophila*, metamorphosis is triggered when larvae attain a target size[25]. Consistent with reduced growth rates upon EcR and HNF4 inhibition in ECs, metamorphosis is significantly delayed under these genetic conditions (Fig. 6e, f and Supplementary Fig. 6d). In conclusion, EcR and HNF4 remodel intestinal function in L3 larvae to accelerate growth and maturation.

## Juvenile remodeling of gut metabolism supports adult fitness

Developmental remodeling of gut function is not only acute, but also transient: intestinal function shuts down as soon as animals enter metamorphosis (Fig. 7a)[8,22]. We sought to determine if, despite its transient nature, this metabolic switch has consequences later in life. To test this possibility, we genetically impaired larval gut remodeling and scored fitness traits in newly eclosed adults. We focused on animals with *Hnf4* silencing in ECs, because inhibition of ecdysone signaling induces lethality during metamorphosis. Adult males with chronic *Hnf4* silencing eclose with 5-13% reductions in systemic protein levels, dry weight and wing area (Fig. 7b–d and Supplementary Fig. 7a).

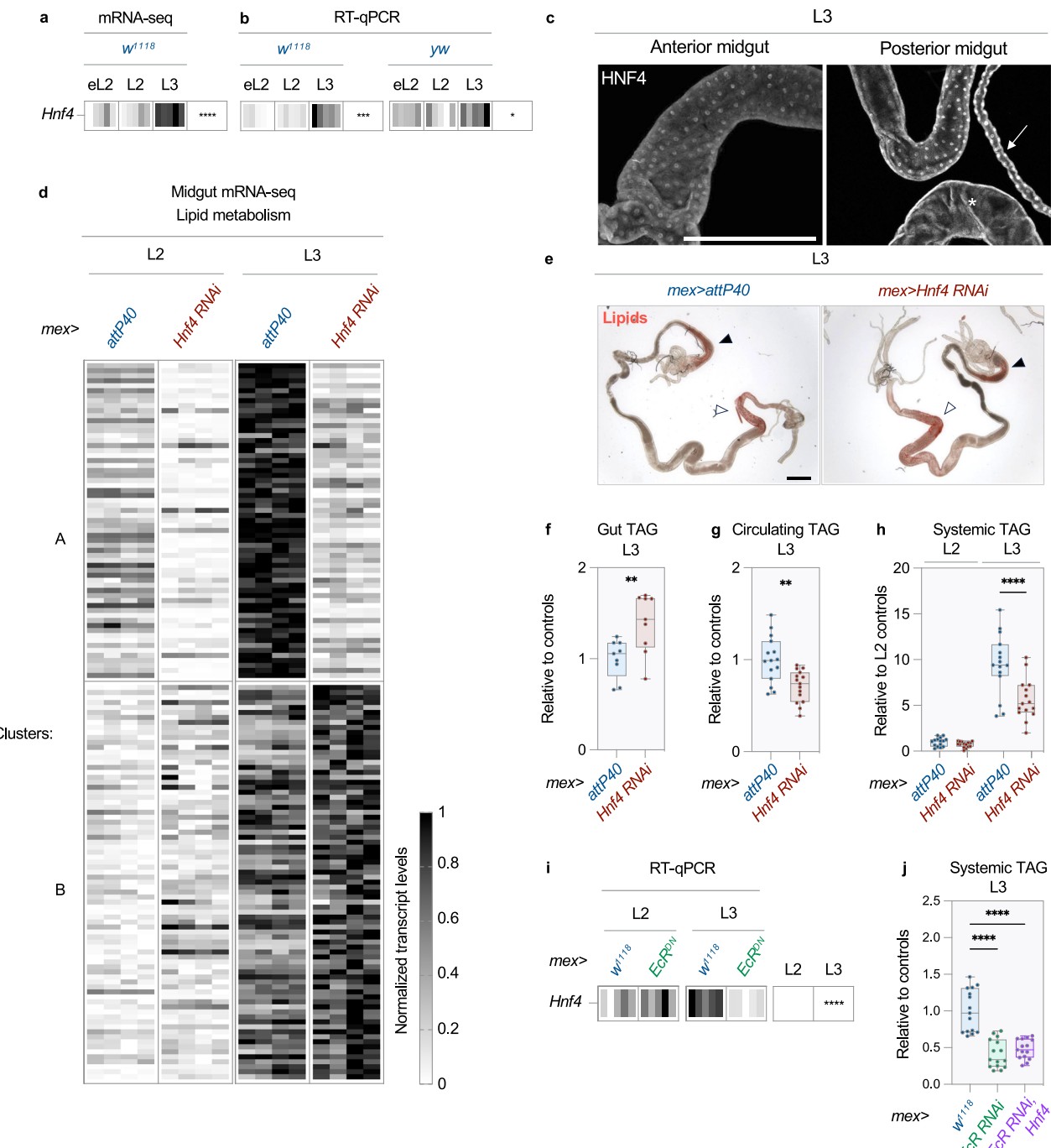

**Fig. 5 | HNF4 is necessary for intestinal metabolic remodeling. a–j** Early L2 (eL2), late L2 (L2) and early L3 (L3) larvae were analyzed. **a, b** *Hnf4* transcript levels in *w¹¹¹⁸* and *yw* midguts. *n* = 5 biological replicates. **c** Anti-HNF4 antibody stains in *mex > w¹¹¹⁸* midguts. Arrow: Malpighian tubules; asterisk: proximal posterior midgut. Representative images from *n* = 30 midguts. Scale bar = 500 μm. (Supplementary Fig. 5a) shows an entire midgut. **d–h** *Hnf4* was silenced in ECs (*mex > Hnf4 RNAi*). *mex > attP40* were used as controls. **d** Hierarchical clustering of developmentally-induced transcripts involved in lipid metabolism (see Supplementary Data 5). *n* = 4 biological replicates. **e** Oil Red O stains of lipids. Closed and open arrowheads indicate anterior and posterior midgut, respectively. Representative images from *n* = 30 midguts. Scale bar = 500 μm. Triglyceride content in **f** midguts (normalized to protein), **g** hemolymph (normalized to volume) and **h** larvae. Values plotted relative to (**f, g**) L3 or (**h**) L2 controls. n = 9, 15 and 15 biological replicates, respectively. **i** *Hnf4* transcript levels in *mex > w¹¹¹⁸* (controls) and *mex > EcRᴰᴺ* midguts. *n* = 5 biological replicates. **j** Triglyceride content in larvae with

*EcR RNAi* without or with *Hnf4* overexpression in ECs (*mex > EcR RNAi* and *mex > EcR RNAi, Hnf4*). Values plotted relative to controls. *n* = 15 biological replicates. **a, b, d, f–j** The contents of replicates are described in the methods. **b, i** One experiment out of three is shown. **c, e, f–h, j** Data from three independent experiments. **a, b, d, i** Transcript levels represented after min-max normalization (see methods). **f–h, j** Dots represent replicates, boxplots extend from the 25th to 75th percentile, whiskers from minimum to maximum, median depicted as a line. **a, b, f–j** Asterisks indicate statistically significant differences (**a, b**) across time-points ((**a**) 1-way ANOVA, **b** Kruskal-Wallis test), **f, g, j** between genotypes ((**f, g**) two-sided Student's *t* test, **j** Kruskal-Wallis test with Dunn's multiple comparisons test) and **h, i** between genotypes at a given stage (2-way ANOVA with Šidák's multiple comparisons). **f** *p* = 0.0078, **g** *p* = 0.0011, **h** *p* = 0.92 (L2) and <0.0001 (L3), **j** *p* < 0.0001 in both comparisons. **p* ≤ 0.05, ***p* ≤ 0.01, ****p* ≤ 0.001, *****p* ≤ 0.0001. Source data are provided as a Source Data file.

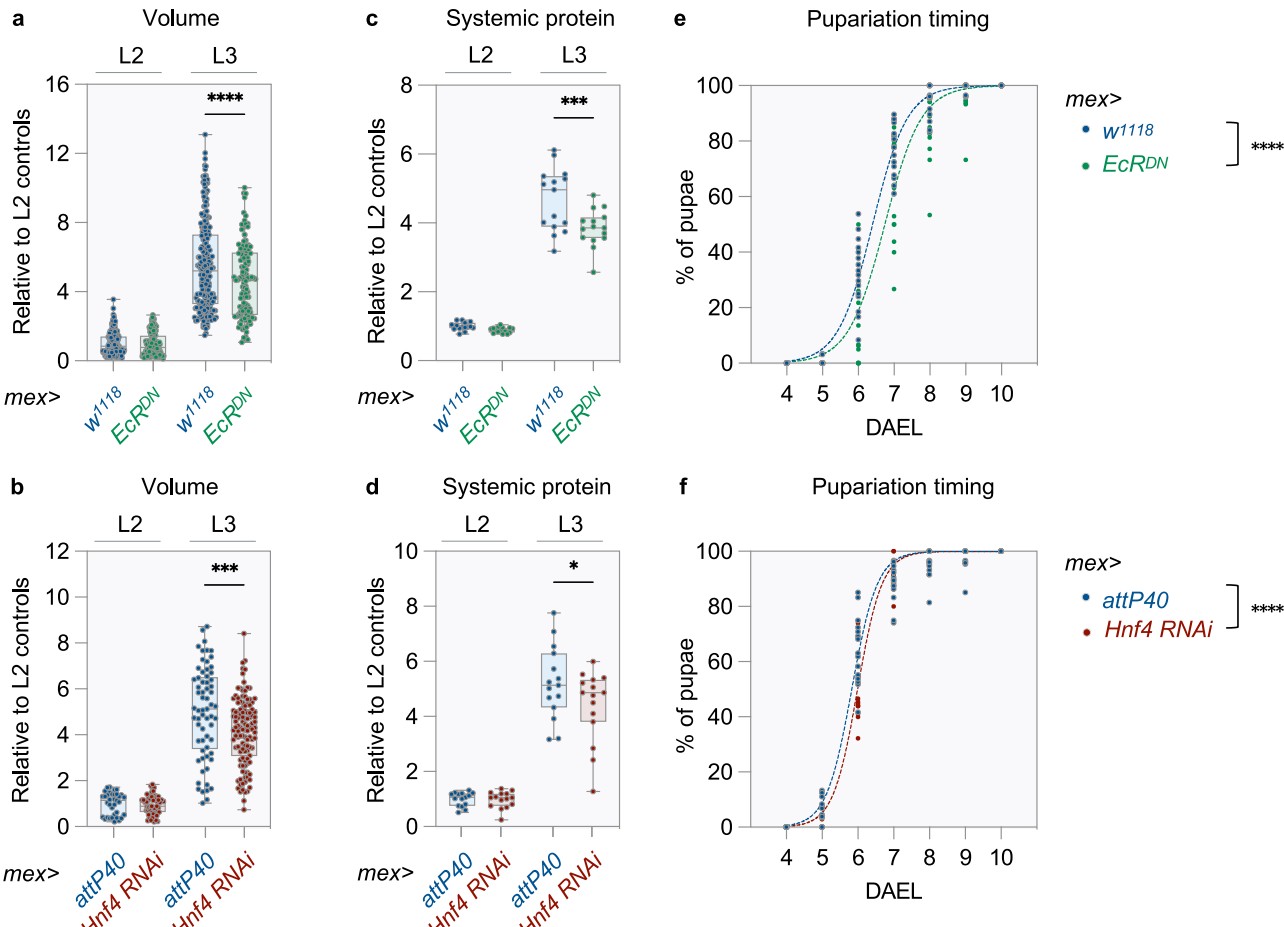

**Fig. 6 | EcR and HNF4 act in ECs to accelerate growth and maturation.**
**a**–**f** Ecdysone signaling or HNF4 were suppressed in ECs by expressing either
(**a**, **c**, **e**) *UAS-EcRDN* or (**b**, **d**, **f**) *UAS-Hnf4 RNAi* with the *mex-GAL4* driver (*mex > EcRDN*
and *mex > Hnf4 RNAi*, respectively). Controls are the progenies of (**a**, **c**, **e**) *mex-GAL4*
and *w1118* parents (*mex > w1118*) or (**b**, **d**, **f**) *mex-GAL4* and *attP40* parents (*mex >
attP40*). **a**, **b** Body volume in L2 and L3 larvae. Data from three independent
experiments. **a** *mex > w1118*: n = 217 (L2) and 215 (L3), mex > *EcRDN*: n = 173 (L2) and
126 (L3). **b** *mex > attP40*: n = 35 (L2) and 65 (L3), *mex > Hnf4 RNAi*: n = 58 (L2) and 118
(L3). **c**, **d** Protein content in L2 and L3 larvae. n = 15 biological replicates, each
containing several animals, from three independent experiments (see methods for
details about the contents of the biological replicates used in these assays).
**e**, **f** Pupariation was scored between D4-10AEL, and is presented as a percentage of
the total number of pupae. The sigmoid curve represents the non-linear least
square regression fit of the data. Median time to pupariation extrapolated from the
non-linear fit analysis: (**e**) *mex > w1118* = 6.40 days and *mex > EcRDN* = 6.74 days, **f**

*mex > attP40* = 5.83 days and *mex > Hnf4 RNAi* = 5.99 days. **e** n = 16 (*mex > w1118*) and
17 (*mex > EcRDN*) biological replicates from three independent experiments and (**f**)
n = 15 biological replicates for both genotypes from two independent experiments
(one vial seeded with 40 embryos considered one biological replicate). **a**–**f** Dots
represent individual biological replicates. **a**–**d** Boxplots extend from the 25th to
75th percentile, whiskers extend from minimum to maximum, median is depicted
as a line. Values plotted relative to L2 controls. **a**–**f** Asterisks indicate statistically
significant difference between (**a**–**d**) genotypes at a given developmental stage (2-
way ANOVA with Šidak's multiple comparisons) or (**e**, **f**) between genotypes (two-
sided extra sum-of-squares F test). **a** p = 0.9658 (L2) and p < 0.0001 (L3), **b**
p = 0.9781 (L2) and p = 0.0003 (L3), (**c**) p = 0.8235 (L2) and p = 0.0002 (L3), **d**
p = 0.9973 (L2) and p = 0.0441 (L3), and **e**, **f** p < 0.0001 for both comparisons.
*p ≤ 0.05, **p ≤ 0.01, ***p ≤ 0.001, ****p ≤ 0.0001. Source data are provided as a
Source Data file.

Therefore, their developmental delay does not fully compensate for
their reduced growth rate, resulting in the emergence of smaller adults
(Figs. 6f and 7b–d). Animals with chronic *Hnf4* silencing in ECs are also
severely lipodystrophic, eclosing with a 47% reduction in body fat
(Fig. 7e). Fat stores acquired during the juvenile phase play a limited
energetic role in the young adult (Supplementary Fig. 7b)[10]. They are
instead used for biosynthetic purposes[10,26,27]. Stored lipids, in parti-
cular, are converted into hydrophobic hydrocarbons in the hours fol-
lowing metamorphosis to waterproof the cuticle and protect against
desiccation[10]. Consistent with this model, animals with *Hnf4* silencing
in ECs eclose without any defect in fluid homeostasis, but are highly
sensitive to desiccation thereafter (Supplementary Fig. 7c and Fig. 7f).
Thus, metabolic remodeling in the larval gut affects the survival of the
young adult in hostile environments. Larval fat also supports oogen-
esis in newly eclosed females[26,28]. Consequently, the size of ovaries
increases rapidly after eclosion in control females, even in the absence

of food (Fig. 7g, h). At eclosion, females with chronic *Hnf4* silencing in
ECs have ovaries that are 14% smaller than those of controls (Fig. 7g, h).
However, this difference increases to 30% the following day in fasted
conditions, which is consistent with the reduced fat stores observed in
this genotype at eclosion (Fig. 7e, g, h). Finally, these females lay fewer
(but equally-fertile) eggs within the first days of adulthood under fed
conditions (Fig. 7i and Supplementary Fig. 7d, e). Thus, transient
remodeling of gut metabolism in the juvenile phase provides young
adults with the metabolic stores necessary to adapt to their new life-
style and to reproduce rapidly. More broadly, these studies shed light
on how transient metabolic shifts in specific organs optimize life his-
tory strategies.

## Discussion
Dietary amino acids are well-established regulators of growth and
maturation in *Drosophila*[20,29–33]. In contrast, other nutrients, including

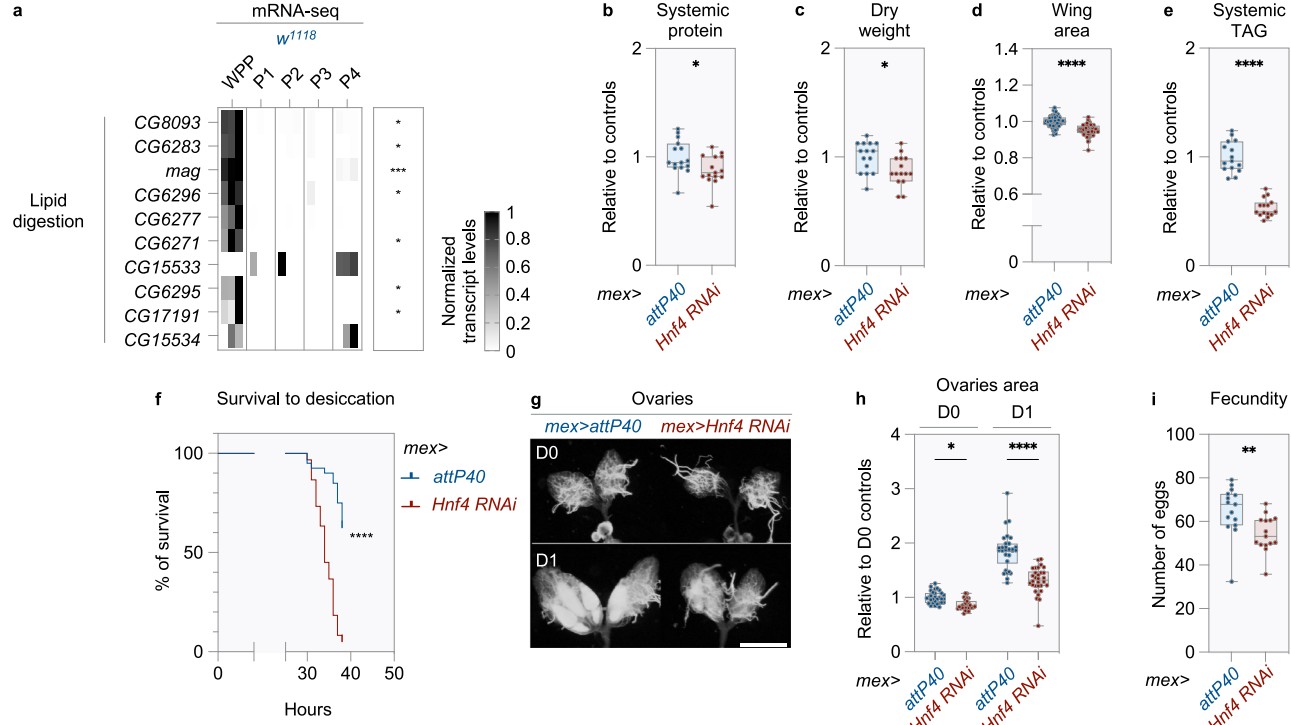

**Fig. 7 | Juvenile remodeling of gut metabolism supports adult fitness. a** Levels of transcripts encoding digestive lipases in $w^{1118}$ animals, from public mRNA-seq data[8]. WPP: white prepupae, P1-P4: first four days of the pupal stage. $n = 3$ biological replicates. Heatmaps depicts transcript levels and statistically significant differences across stages. Transcript levels are represented using a greyscale after min-max normalization, see methods for details. **b–i** *Hnf4* was silenced in ECs by expressing *UAS-Hnf4 RNAi* with the *mex-GAL4* driver (*mex>Hnf4 RNAi*). *mex>attP40* animals were used as controls. **b** Protein, **c** dry weight, **d** wing area and **e** total triglyceride were scored in newly eclosed adult males. **b–e** Values are plotted relative to controls. Data from three independent experiments. **b, c, e** $n = 15$ biological replicates, each containing several specimens (see methods for details), **d** $n = 60$ wings. **f** Survival to desiccation in newly eclosed males. Survival was monitored for 38 hours. Median lifespan: *mex>attP40*: >38 h and *mex>Hnf4 RNAi*: 34 hours. *mex>attP40*: $n = 40$ and *mex>Hnf4 RNAi*: $n = 60$ animals. One representative experiment out of three is shown. **g, h** Ovaries from newly eclosed

females and one-day-old females subjected to mating and wet starvation. **g** representative images, scale bar = 500 μm. **h** quantification of the two-dimensional area of ovaries from images. **g, h** $n = 30$ specimens from three independent experiments. **i** Fecundity over the first five days of adult life under fed conditions. $n = 15$ biological replicates from three independent experiments. Eggs laid by a group of five females are considered one biological replicate. **b–e, h, i** Dots represent biological replicates or individual organs, boxplots extend from the 25th to 75th percentile, whiskers from minimum to maximum, median depicted as a line. **a–f, h, i** Asterisks indicate a statistically significant difference: (**a**) across timepoints (Kruskal-Wallis test), (**b–f, i** between genotypes ((**b–e**) two-sided Student's *t* test, **f** two-sided Mantel-Cox test and (**i**) two-sided Mann-Whitney test), or (**h**) between genotypes at a given stage (2-way ANOVA with Šidak's multiple comparisons). **b** $p = 0.0321$, **c** $p = 0.0201$, **d–f** $p < 0.0001$, (**h**) $p = 0.0411$ (D0) and $p < 0.0001$ (D1), **i** $p = 0.0023$. *$p \leq 0.05$, **$p \leq 0.01$, ***$p \leq 0.001$, ****$p \leq 0.0001$. Source data are provided as a Source Data file.

lipids, have received far less attention. *Drosophila* are auxotrophs for sterols, which are essential for cellular membranes and hormone synthesis, and these insects cannot synthesize several polyunsaturated fatty acids[19,20,34]. Gut metabolic remodeling, with increased lipid digestion and export, could accelerate systemic growth by supplying peripheral tissues with these essential nutrients[19]. It may also support systemic anabolism in a less direct manner. For instance, this metabolic switch could directly enhance the function of ECs: elevated intracellular lipolysis and beta-oxidation could fulfill the energetic demands of increased nutrient metabolism and export, while remodeling of membrane lipids could promote nutrient trafficking. This metabolic switch could also promote growth by altering the function of peripheral organs. Adipose tissue secretes adipokines that regulate the secretory function of insulin-producing cells, thereby affecting growth[30–32,35–37]. As gut metabolic remodeling supports rapid fat storage during the L3 stage, it is likely to increase the size, and therefore the endocrine capacity, of the fat body during this period (Figs. 4h, 5h and Supplementary Fig. 4b). In parallel, inhibiting gut metabolic remodeling reduces, but does not halt, the acceleration of growth after the L2 stage (Fig. 6a–d and Supplementary Fig. 6b). This suggests the existence of additional physiological adaptations supporting this growth spurt. However, disturbing gut metabolic remodeling has long-term consequences,

resulting in smaller adults with reduced fitness (Fig. 7b–i). Finally, these effects were observed using a nutrient-rich diet. Gut metabolic remodeling may have a more pronounced impact on growth and fitness in suboptimal nutritional conditions, such as those encountered in the wild.

Systemic fat stores and growth are not restored when *Hnf4* is overexpressed in animals with inhibition of ecdysone signaling (Fig. 5j and Supplementary Fig. 6c). This could be explained by HNF4 controlling part of enterocyte metabolic remodeling downstream of EcR. Accordingly, several transcripts involved in lipid metabolism require EcR, but not HNF4, for induction at the L3 stage (Fig. 4d and Supplementary Fig. 5b). This is the case for *mag*, *CG6271* and *CG6277*, which encode digestive lipases (Fig. 4d and Supplementary Fig. 5b). Therefore, *Hnf4* overexpression may not rescue the digestive defects resulting from the inhibition of ecdysone signaling, and thus the absorption of dietary fats. This would explain why *Hnf4* overexpression cannot restore adiposity when ecdysone signaling is inhibited in ECs (Fig. 5j). Conversely, HNF4 is necessary for lipid export from the intestine[14], explaining why its silencing induces intestinal steatosis and reduces adiposity, recapitulating hallmarks of EcR inhibition (Figs. 4e–h and 5e–h). Taken together, these observations support a model in which EcR acts through several pathways, including HNF4 signaling, to induce intestinal metabolic remodeling. This

metabolic switch is required in its entirety to support the buildup of fat stores and growth acceleration.

Ecdysone is released in repeated pulses during larval development, but it triggers gut metabolic remodeling only in the last larval instar (Figs. 2a and 4d)[7]. Therefore, additional factors may act alongside this hormone to control this metabolic transition. In most insects, juvenile hormone (JH) inhibits the metamorphic effect of ecdysone in the early larval stages[38]. JH levels drop rapidly during the last larval stage, allowing metamorphosis to proceed[39,40]. Our data suggest that a similar mechanism gatekeeps gut metabolic remodeling. The induction of intestinal lipid metabolism in L3 coincides with increased expression of *broad*, a member of the metamorphic gene network, suggesting inhibition of JH signaling (Fig. 4a, b, d). Similarly, we observed a sharp decrease in transcripts related to ribosomal biogenesis, a process promoted by JH[41] (Fig. 2a, b). An epistatic relationship between JH, EcR, HNF4 and lipid metabolism has also been documented in adult mosquitoes[15]. Therefore, JH is likely to act alongside ecdysone to control the timing of gut metabolic remodeling. Finally, environmental factors, including nutrients and commensal bacteria, can also affect hormonal signaling and gut metabolism[33,42,43]. These additional inputs may refine the timing, duration, and magnitude of gut metabolic remodeling, thereby influencing its overall impact on growth and fitness.

Although nutrient absorption plays a fundamental role in growth, it is only fully activated midway through the juvenile phase (Fig. 3b, d and Supplementary Fig. 2c, d). This paradox suggests a trade-off between maximal intestinal function and juvenile health. In the wild, larvae feed on decaying plant matter and ingest a wide variety of microbes[22]. Nutrient release by digestive enzymes fuels not only the host, but also commensal bacteria, and could facilitate dysbiosis or the establishment of pathogens[44]. Accordingly, gut metabolic remodeling coincides with a 100-fold increase in intestinal bacterial density and immune activation between the L2 and L3 stages[33,45,46]. Adding to this risk, digestive enzymes can leak into the circulation, leading to multiorgan failure by degrading membrane receptors, plasma proteins and signaling lipids[47–49]. Meanwhile, the midguts of L1 and L2 larvae have limited regenerative capability, as this process relies on a pool of progenitors that expands during the third larval instar[50]. Therefore, digestion may only be fully activated once the juvenile gut has acquired sufficient regenerative capability to withstand potential negative side effects.

Consistent with the role of ecdysone in inhibiting Myc, a master regulator of ribosomal biogenesis[51], this process is suppressed in L3 midguts (Fig. 2a, b). Although ribosomes are essential for growth[52], the midgut still grows between the L2 and L3 stages (Supplementary Fig. 3b, c, e, f). This could be explained by the long half-life of these ribonucleoprotein particles, which can exceed several days[53]. Ribosomes may be actively produced during the L1 and L2 stages to meet the needs of ECs during the final larval stage, which lasts approximately 48 h (Fig. 1a, b). The suppression of ribosome biogenesis would then allow resources to be reallocated to EC-specific functions, such as nutrient metabolism and export, ultimately supporting systemic growth. Mammalian intestinal epithelial cells also display a reduction in MYC targets when they differentiate along the crypt-villus axis[54], further supporting the existence of a trade-off between ribosome biogenesis and specialized cell functions. The genetic tractability of *Drosophila* would be a valuable asset to validate the existence of this trade-off, and how it affects systemic metabolism, growth and fitness.

From a broader perspective, this juvenile switch in intestinal metabolism exemplifies the remarkable plasticity of the gut during key life events, such as weaning or pregnancy[55–59]. From a metabolic point of view, the L3 midgut resembles the neonatal mammalian intestine, which is specialized for lipid-rich nutrition - milk providing up to 60% of calories in the form of fat[56–59]. In mammals, the transcriptional repressor Blimp1/PRDM1 maintains this neonatal state by suppressing the adult metabolic program[56–58]. Blimp-1 is regulated by ecdysone signaling in *Drosophila*[60] and has many putative binding sites in proximity to genes downregulated in L3 midguts (Supplementary Data 4). Therefore, it may act alongside HNF4 to reconfigure intestinal metabolism during development. Similarly, the mammalian ortholog of EcR, Liver X Receptor, controls intestinal lipid metabolism and transport[61]. Finally, the changes in lipid metabolism and ribosomal function observed after entry into the L3 stage mirror EC maturation from crypts to villi in mammals, a process driven by HNF4A/G (Fig. 2a, b)[54,62]. Therefore, the evolutionary conservation of these pathways positions *Drosophila* as a powerful model for understanding organ plasticity during development, and for linking early-life metabolic states to longer-term physiological outcomes (Fig. 7b–i). More broadly, by uncovering a transient gut metabolic program that shapes juvenile growth, maturation and adult fitness, this study provides a foundation for understanding how early metabolic states can predispose to lifelong physiological health or disease.

## Methods

### *Drosophila* strains and handling
A detailed list of the strains used in these studies is provided in (Supplementary Table 1). Stocks were routinely maintained at room temperature on a standard diet (SD, see recipe below). For crosses and experiments, animals were kept at 25 °C with a 12-hour light/dark cycle (light cycle: 7a.m.–7p.m.) on a yeast-only diet (YOD, see recipe below). We used a yeast diet with no added sugars, as these nutrients suppress digestive activities[42,43]. See the section "Synchronous egg-laying" below for more details on genetic crosses. For transgene expression in ECs, we used the *mex-GAL4* driver line. For experiments involving conditional and ubiquitous transgene expression, we used the thermosensitive *tub-GAL4, tub-GAL80^{ts}* driver line. In this case, crosses were made at 22 °C and progenies transferred at 29 °C from 48 hours after egg laying to induce transgene expression.

### *Drosophila* strains were fed the following diets
SD: 160 g of agar were added in 20 L of boiling water. After complete dissolution, 360 g of active dried yeast, 1.6 kg of corn flour and 200 g of soy flour were mixed and incorporated into the solution. 1.2 L of malt extract was mixed with 330 mL of beet syrup, added to the solution and the mixture was cooked for 1 hour at 86 °C. After 1 hour, 4 L of water were added and the mixture was let to cool down to 60–63 °C. Finally, 130 mL of nipagin (15% w/v, dissolved in ethanol) and 130 mL of propionic acid were added before the food is poured into vials and bottles.

YOD: 80 g active dry yeast (Bäckerei Spiegelhauer, Cat#1278) and 10 g agar (Fisher Scientific, Cat#BP1423-2) were mixed with 1 L water and boiled on a magnetic hotplate stirrer for 10 min. After cooling below 60 °C, 4 mL of 99.5% propionic acid (Sigma-Aldrich, Cat#81910-1 L) and 5.2 g of methyl 4-hydroxybenzoate sodium salt (VWR, Cat#235145000) were added as preservatives. The diet was then manually poured into vials or bottles.

YOD with blue dye: same recipe as above, 8 g of Brilliant Blue FCF (Carl Roth, Cat#2981.3) was added to 1 L of YOD at the same time as preservatives.

### Synchronous egg-laying
Synchronous egg-laying was performed for all experiments involving larvae or pupae. For this, 100 virgin females and 50 males aged 3–14 days were crossed in cages made out of polypropylene bottles (Kisker Biotech, Cat#789022B) with 60 mm Petri dishes (VWR, Cat#734-2794) filled with YOD at their bottom. After a 24-h acclimation period, the spent Petri dishes were replaced with fresh ones for overnight egg-laying (5 p.m.–8a.m.). The following morning, batches of 40 embryos were collected using a scalpel and transferred to

28.5 × 95 mm polystyrene vials (Kisker Biotech, Cat#789009) containing YOD and closed with a cellulose acetate stopper (Kisker Biotech, Cat#789035). Animals were incubated at the temperatures described above until collection.

## Larval staging and sizing

Larvae were staged according to the morphology of their anterior spiracles[63]. One vial seeded with 40 embryos was considered as one biological replicate. For sizing, animals were collected in PBS and quickly incubated on a hot plate until they ceased to move. Larvae were then quickly rinsed in distilled water, transferred to a glass slide and imaged by bright-field microscopy using a SZX16 stereomicroscope (Olympus) with a DP72 digital microscopy camera (Olympus) and a U-TV0.63XC Camera Adapter (Olympus). Images were acquired with the cellSens Standard Software (Olympus, v1.11). Length was measured from the anterior limit of the mouth hooks to the posterior part of the spiracles and width was measured at the midpoint of the body using Fiji (v.1.54p, RRID:SCR_002285). Body volume ($V$) was calculated using the formula for an ellipsoid ($V = \frac{4\pi}{3} * \frac{length}{2} * (\frac{width}{2})^2$), as previously described[17].

## Imaging after blue dye feeding

Larvae were fed YOD with blue dye and collected and staged as described above. Instead of heat treatment, they were briefly put on ice for immobilization. Larvae were then quickly transferred to a glass slide and imaged using bright-field microscopy with a SZX16 stereomicroscope (Olympus) equipped with a DP72 digital microscopy camera (Olympus) and a U-TV0.63XC Camera Adapter (Olympus). Images were acquired with the cellSens Standard Software (Olympus, v1.11). Two independent experiments were performed, each with $n = 30$ larvae originating from vials seeded with 40 embryos.

## Pupariation timing

Crosses were performed as described above and pupae counted daily from day 4 until day 10 after egg laying. One vial seeded with 40 embryos was considered as one biological replicate. The pupariation percentage was calculated by dividing the cumulative number of pupae by the total number of pupae in a given vial at D10AEL. Logistic regression, with the bottom and top values constrained to 0 and 100 respectively, was performed to determine the median time to pupariation for each genotype.

## Adult collection

For experiments involving newly eclosed adults, 50 virgin females and 25 males aged 3–14 days were crossed in polypropylene bottles (Kisker Biotech, Cat#789022B) containing YOD and closed with a cellulose acetate stopper (Kisker Biotech, Cat#789034). Parents were transferred to fresh bottles containing YOD every two to three days, and the bottles containing their progenies were kept at 25 °C. Bottles were then examined daily to collect newly eclosed adults. Newly eclosed adults were anesthetized using $CO_2$ and identified by the presence of meconium in their abdomens. They were handled according to the methods described below.

## Body weight and water content

Newly eclosed males were grouped into sets of five to make up one biological replicate. These were placed in 1.5 mL pre-weighed microcentrifuge tubes (TH. Geyer, Cat#7696753) and kept at −20 °C until further processing. One empty tube per set allowed to score tube weight variation. Once all the samples had been collected, the tubes were placed at 22 °C for one hour to allow them to reach equilibrium temperature, after which they were weighed on a precision scale (Avantor, Cat#611-4797) to determine sample wet weight (WW). Carcasses were then dried by placing open tubes at 37 °C for 48 hours. Tubes were then transferred at 22 °C for one hour before being closed and weighed again to determine sample dry weight (DW). Water

percentage (WP) was calculated for each sample using the following formula: WP = 100*(WW-DW)/WW.

## Wing size

Twenty newly eclosed animals were anesthetized using $CO_2$. Their left wings were dissected and transferred to a glass dish containing 70% ethanol. Wings were then transferred onto a glass slide using a P1000 micropipette with a cut tip. Most of the liquid was removed, and wings repositioned using fine forceps. A 1:1 mixture of glycerol and ethanol was added, and then a coverslip placed on top. Wings were imaged by bright-field microscopy using a SZX16 stereomicroscope (Olympus) with a DP72 digital microscopy camera (Olympus) and a U-TV0.63XC Camera Adapter (Olympus). Images were acquired with the cellSens Standard Software (Olympus, v1.11). Wing area was measured using Fiji (v.1.54p, RRID:SCR_002285), following the landmarks shown in (Supplementary Fig. 7a).

## Resistance to desiccation

Newly eclosed adult males were collected at 9 a.m. and transferred to 28.5 × 95 mm empty polystyrene vials (Kisker Biotech, Cat#789009) closed with a cellulose acetate stopper (Kisker Biotech, Cat#789035) at a density of 10 animals per vial and put at 25 °C. Lethality was scored at 0, 4, 8, 24, and 28 h, then hourly from 30 to 38 h. Data from animals collected in the same week and scored at the same timepoints were pooled as one independent experiment.

## Resistance to starvation

Newly eclosed adult males were collected and transferred to 28.5×95 mm polystyrene vials (Kisker Biotech, Cat#789009) containing a cellulose acetate stopper (Kisker Biotech, Cat#789035) saturated with tap water, and at a density of 10 animals per vial. The vials were sealed with a second cellulose acetate stopper and kept at 25 °C. Animals were transferred to a new starvation vial twice a week. Lethality was scored daily until all the animals were dead. Data from animals collected in the same week and scored at the same timepoints were pooled as one independent experiment.

## Ovary size

Ten newly eclosed females were collected in the morning and transferred to 28.5×95 mm polystyrene vials (Kisker Biotech, Cat#789009). The vials were left empty and frozen directly at −20 °C for D0 measurements. For D1 measurements, females were transferred to starvation vials (see section above) with five $w^{1118}$ males to allow mating. After 24 h, vials were frozen and stored upside down at −20 °C until dissection. Animals were then thawed in PBS, their ovaries dissected, transferred to a glass slide, and imaged using bright-field microscopy with a SZX16 stereomicroscope (Olympus) equipped with a DP72 digital microscopy camera (Olympus) and a U-TV0.63XC Camera Adapter (Olympus). Images were acquired with the cellSens Standard Software (Olympus, v1.11). The two-dimensional area of the two ovaries, the lateral oviducts and the junction with the common oviduct was measured using Fiji (v.1.54p, RRID:SCR_002285). For illustration, an independent set of ovaries was imaged immersed in PBS in a dissecting dish to improve visualization of the structure. A representative pair of ovaries were selected for the (Fig. 7g).

## Fecundity and fertility

Five newly eclosed females were collected in the morning and transferred to 28.5 × 95 mm polystyrene vials (Kisker Biotech, Cat#789009) containing YOD along with three $w^{1118}$ males. The vials were closed with a cellulose acetate stopper (Kisker Biotech, Cat#789035). The next day, adults were transferred to new vials without $CO_2$. This procedure was repeated daily from day 2 until day 6 after eclosion. Dead males were replaced throughout the experiment to maintain the male-to-female ratio in each vial. Fecundity was assessed by counting the number of eggs

and dividing it by the number of live females in each vial daily. Fertility was assessed by recording the proportion of hatched eggs 48 h after egg laying. The survival of progenies was calculated by dividing the number of pupae at 10DAEL by the number of eggs that successfully hatched in a given vial. For all fecundity and fertility-related experiments, one vial was considered a biological replicate.

## Sample processing for metabolite quantification

Whole animals or tissues were collected in 1.5 mL Safe-Lock microcentrifuge tubes (Eppendorf, Cat#0030123611) and snap-frozen in liquid nitrogen. Samples were then stored at −80 °C until further analysis. For metabolite quantification in whole larvae, thirty L1, fifteen early L2, five L2, three L3 and two late L3 larvae were collected at D1, D2, D3, D4, and D5, respectively, to make up one biological replicate. They were staged, rinsed in PBS, dried on a Kim wipe, and snap-frozen. For quantifications in adults, five newly eclosed males were beheaded and pooled to make up one biological replicate, before being snap-frozen. For midgut analysis, thirty L2 or five L3 midguts without hindgut and Malpighian tubules were collected at D3 or D4, respectively, rinsed and snap-frozen, making up one biological replicate. For hemolymph analysis, forty L2 or ten L3 larvae were collected at D3 or D4, respectively, rinsed in PBS, and dried on a Kim wipe. They were then deposited on a glass slide on a 90 mm Petri dish (VWR, Cat#391-0556 P) containing ice wrapped in Parafilm (Heathrow Scientific, PM999). These larvae were then bled by tearing their cuticle open. 2 μL of hemolymph were collected using a micropipette to make up one biological replicate. Hemolymph samples were collected and snap frozen within 2 min. On the day of the metabolite assays, samples were kept on dry ice until being processed. Samples containing whole animals and tissues were homogenized individually using a motor and pestle (DWK Life Sciences, Cat#K749540-0000) in ice-cold PBS (120 μL for whole larvae and adults, 80 μL for midguts) for 15 s. Hemolymph was directly diluted in 48 μL ice-cold PBS. 10 μL were collected for protein quantification for all samples except for hemolymph. Each sample was then directly heat-treated in a water bath for 10 min at 70 °C to inactivate endogenous enzymes and prevent metabolite degradation.

## Protein quantification

Protein levels were determined using Protein Assay Dye Reagent (Bio-Rad, Cat#5000006, diluted 1:4 in MilliQ water (v/v)). The following dilutions in PBS (v/v) were used for whole-animal and tissue homogenates: 1/4 for D1 and D2 larvae; 1/8 for D3 larvae; 1/10 for D4 and D5 larvae and adults; 1/5 for D3 and D4 midguts; 1/7 for enzymatic activity in D3 midguts; 1/6.25 for enzymatic activity in D4 midguts. 10 μL of diluted homogenates were transferred to a 96-well plate (Avantor, Cat#43001-0119) and incubated with 200 μL of diluted Protein Assay Dye Reagent. The plate was covered with Parafilm (Heathrow Scientific, Cat#PM999) and incubated for 5 min at 37 °C before absorbance was read at 595 nm using a Multiscan SkyHigh microplate spectrophotometer and the SkanIt software (v5.0, Thermo Fisher Scientific). Protein concentration in homogenates was determined using a standard curve generated with diluted BSA samples (VWR, Cat#1.12018.0025, diluted in PBS) that had been treated with Bio-Rad Protein Assay Dye.

## Triglyceride quantification

To determine triglyceride concentration, 20 μL of heat-treated homogenates was incubated with either 20 μL of PBS ("untreated samples") or 20 μL of triglyceride reagent (Sigma Aldrich, Cat#T2449-10ML). Pure lysates were used for all reactions, except for D4 and D5 larvae samples, for which lysates were diluted 1/8 and 1/10 (v/v) in PBS, respectively, for total glycerol determination. Samples were then incubated for 1 h at 37 °C, after which they were centrifuged at maximum speed for 3 min at 4 °C. Then, 30 μL of the supernatant was incubated with 100 μL of free glycerol reagent (Sigma Aldrich, Cat#F6428-40ML) for 5 min at 37 °C in a 96-well plate (Avantor, Cat#43001-0119) covered with Parafilm (Heathrow Scientific, Cat#PM999) to determine the free glycerol concentration. Absorbance was then read at 540 nm using a Multiscan SkyHigh microplate spectrophotometer and the SkanIt software (v5.0, Thermo Fisher Scientific). The absorbance of untreated samples was then subtracted from that of samples incubated with triglyceride reagent. This value was then used to calculate the triglyceride content of each sample based on a standard curve of serial dilutions of a triolein-equivalent (triolein-equivalent glycerol standard 2.5 mg/mL, Sigma-Aldrich, CAT#G7793) treated with the free glycerol reagent. Triglyceride levels were normalized to protein when indicated in the legends.

## Glycogen quantification

To measure whole body glycogen levels, heat-treated homogenates were centrifuged at maximum speed for 3 min at 4 °C. 15 μL of the supernatant was transferred to a 96-well plate and incubated with either 15 μL of PBS ("untreated" samples), or 15 μL of amyloglucosidase diluted in PBS (Sigma Aldrich, Cat#A1602-25MG, 1.5 μL of amyloglucosidase stock solution diluted in 1 mL PBS). Pure lysate was used for D1-D3 larvae, while D4-D5 larval homogenates were diluted 1/2 (v/v) in PBS prior incubation with amyloglucosidase. Samples were incubated for 1 h at 37 °C. Then, 15 μL were incubated with 100 μL hexokinase reagent (Sigma Aldrich, Cat#GAHK20) for 2 h at 22 °C in a 96-well plate (Avantor, Cat#43001-0119) covered with Parafilm (Heathrow Scientific, Cat#PM999). Absorbance was then read at 340 nm using a Multiscan SkyHigh microplate spectrophotometer and the SkanIt software (v5.0, Thermo Fisher Scientific). To determine glycogen levels, the absorbance measured for free glucose in the untreated samples was subtracted from that of the absorbance of the samples digested with amyloglucosidase. The glycogen content of each sample was then calculated based on a standard curve with serial dilutions of glycogen (Sigma-Aldrich, G0885) digested with amyloglucosidase and treated with the hexokinase reagent as described above. To determine the free glucose level, the absorbance measured for free glucose in the untreated samples was calculated based on a standard curve with serial dilutions of glucose treated with the hexokinase reagent.

## Enzymatic activity

For the determination of amylase activity, 20 L2 or 10 L3 midguts were dissected on ice to make up one biological replicate. For the determination of lipase activity, 20 L2 or five L3 midguts were dissected on ice to make up one biological replicate. Midguts were snap-frozen in microcentrifuge tubes (VWR, Cat#432-0351) containing 100 μL of sterile 1 mm glass beads (Carl Roth, Cat#N031.1) and stored at −80 °C until further processing. Amylase Activity assay kit (Sigma Aldrich, Cat#MAK009-1KT) and Lipase Activity assay kit (Sigma Aldrich, Cat#MAK047) were used for the quantification of the respective enzymatic activities. On the day of the assay, samples were homogenized in 70 μL of the respective assay buffers with a Precellys 24 homogenizer (Bertin Instruments) for 15 second at 5000 rpm. Diluted lysates were then transferred to 2 mL Eppendorf tubes and further diluted 1/2.75 or 1/10 (v/v) in the respective assay buffers for the L2 and L3 timepoints, respectively. Samples were then centrifuged at 13.000 × $g$ for 10 min at 4 °C. 10 μL of supernatant were used for protein quantification as described above. 50 μL of supernatant was transferred to a 96-well plate (Avantor, Cat#43001-0119) and processed further with the kits according to the manufacturer's recommendations. Enzymatic activities were normalized to protein levels to take into account variations in tissue size.

## RNA extraction

For RT-qPCR experiments, total RNA was extracted from 10, 5, or 3 midguts at D2, D3 and D4, respectively, for $w^{1118}$ and $yw$ larvae, to make

up one biological replicate. For all the other genotypes, RNA was extracted from 10 and 3 midguts at D3 and D4, respectively, to make up one biological replicate. For mRNA-seq experiments, total RNA was extracted from 60, 30 or 10 midguts that were dissected on ice at D2, D3 and D4, respectively, to make up one biological replicate. All samples were snap-frozen in liquid nitrogen in microcentrifuge tubes (VWR, Cat#432-0351) with 100 μL of sterile 1 mm glass beads (Carl Roth, Cat#N031.1) and stored at −80 °C until further processing. Total RNA was extracted using the Reliaprep RNA Tissue Miniprep System (Promega, Cat#Z6012). 500 μL of lysis buffer was added to the microtubes and tissues were homogenized using a Precellys 24 homogenizer (Bertin Instruments) at 5,000 rpm for 15 seconds. An additional centrifugation step at $1000 \times g$ for 1 min was performed to eliminate the foam formed after tissue homogenization. Subsequent RNA extraction steps were performed according to the manufacturer's recommendations, with RNA eluted in 20 μL of nuclease-free water (Promega, Cat#P119E).

## cDNA synthesis and RT-qPCR

cDNA synthesis was performed using 0.25-1 μg RNA and the qScript cDNA Synthesis Kit (Quantabio, Cat#733-1178), following the manufacturer's protocol. cDNA were then diluted 1/10 (v/v) in nuclease-free water (Promega, Cat#A6001). RT-qPCR reactions were performed on cDNA using GoTaq qPCR Master Mix (Promega, Cat#A6001), on a QuantStudio 3 or 5 (Applied Biosystems) with accompanying software. Oligonucleotides (Integrated DNA Technologies) were used at 0.25 μM and are listed in (Supplementary Table 2). Fold changes were determined using the ΔΔCt method. Transcript levels were normalized to the arithmetic mean of the Ct values of the two reference transcripts *rp49* (also known as *RpL32*) and *tbp*.

## mRNA-seq library preparation

The Cologne Center for Genomics performed the library preparation and sequencing. The Stranded Truseq RNA Sample Preparation Kit (Illumina) was used to prepare the libraries from 1 μg of total RNA. The ERCC RNA Spike-In Mix (Thermo Fisher Scientific, Cat#4456740) was added to the samples before library preparation. Polyadenylated mRNA was selected using poly-T oligo-attached magnetic beads. Then, it was purified and fragmented with divalent cations at a high temperature. Random primers were then used to reverse transcribe the RNA fragments. Second-strand cDNA synthesis was performed using RNase H and DNA polymerase I, after which indexing adapters were ligated following end repair and A-tailing. Fifteen PCR cycles were performed to amplify the products and generate the cDNA libraries. Equimolar amounts of the libraries were pooled after validation and quantification (Agilent Tape Station). The pooled libraries were quantified with the Peqlab KAPA Library Quantification Kit (Roche) and the 7900HT Sequence Detection System (Applied Biosystems, Thermo Fisher Scientific) and sequenced using a 100 bp paired-end protocol with a NovaSeq 6000 sequencing system (Illumina).

## Analysis of mRNA-seq data

All details related to quality control, adapter trimming, read filtering, alignment to the *Drosophila* genome, transcript quantification and differential expression analysis are provided with the raw mRNA-seq data on the GEO database, accession numbers GSE283125 and GSE315710.

For the analyses presented in (Fig. 2), differential expression analysis was performed on filtered genes with ≥ 0.5 CPM in ≥ 1 library in the GSE283125 dataset using the iDEP pipeline (v0.96)[64] (http://bioinformatics.sdstate.edu/idep96/) containing the DESeq2 package[65] (RRID:SCR_015687). The 1000 most variable transcripts throughout all samples were subjected to a hierarchical clustering analysis and GO term enrichment analysis was performed on the identified clusters using the iDEP pipeline (v0.96)[64].

For the analysis of the GSE315710 dataset presented in (Fig. 5d), we used GO-Term-Finder (v0.86, https://go.princeton.edu/cgi-bin/GOTermFinder) to identify transcripts associated with the gene ontology term "lipid metabolic process" among all the transcripts that are significantly upregulated in control midguts between the L2 and L3 stages (log2FC ≥ 0.5, $p_{adj}$ ≤ 0.05). The transcripts were subjected to a min-max normalization based on CPM values across all samples. Hierarchical clustering was performed using the pheatmap R package (v1.0.13) with Euclidean distance as the clustering method. In both cases, heatmap visualization was done using GraphPad Prism (v10, RRID:SCR_002798)

## Identification of digestive enzymes

We used the FlyAtlas2 database[66] to identify genes coding for digestive enzymes. Out of the 17158 referenced genes, 418 are "larval midgut-specific genes" with a midgut enrichment ($FPKM_{organ}$/$FPKM_{whole\ body}$) > 5-fold higher than in the other organs. To identify genes encoding digestive enzymes in this list, we collected information relating to their associated GO terms Biological Process (BP), Molecular Function (MF) and Cellular Component (CC) using the Batch Download tool on FlyBase[67] (http://flybase.org/batchdownload). Candidate genes were then manually curated by selecting those for which the CC-GO term mentioned "extracellular", the BP-GO term mentioned "metabolic" or "catabolic", and the MF-GO term indicated enzymatic catabolic activity. Note that genes encoding maltases, beta-galactosidase, Jonah proteases and trypsins do not fully match these criteria, but were added to the list due to their likely involvement in digestion. The final list is available in (Supplementary Data 3).

## TF binding site analysis

We used Cytoscape[68] (v.3.10.1, RRID:SCR_003032) with the iRegulon plugin[69] (v1.3, https://apps.cytoscape.org/apps/iregulon) to analyze the enrichment of putative TFs binding sites in the vicinity of genes included in clusters A, B, D and E (Fig. 2a). (Supplementary Data 4) list the parameters used for the analysis and show for each motif the matching candidate TFs and the target genes containing this motif in their vicinity (5 kb upstream and full transcript).

## Analysis of public ChIP-seq data

Publicly available ChIP-seq data were analyzed via the TF2TG online portal (https://www.flyrnai.org/tools/tf2tg/web/)[24]. Using this portal, we identified peak locations of ecdysone-related TFs within 5 kb of the transcription start site of the *Hnf4* locus. These analyses and the reference to the source datasets are detailed in (Supplementary Data 6).

## Tissue fixation and staining

For (immuno)histochemistry, larval midguts were dissected in PBS and fixed in 4% paraformaldehyde (Thermo Scientific, Cat#43368) for 16 h at 4 °C and rinsed thrice in PBS. For immunostains, tissues were permeabilized and blocked for 8 hours at 4 °C by incubating them in a blocking buffer containing normal donkey serum (Biozol, Cat#LIN-END9010-10) diluted 1/20 (v/v) in PBS with 0.5% Triton-X100 (Merck, Cat#1,086,031,000) (PBST). Samples were incubated in primary antibodies diluted in blocking buffer for 16 h at 4 °C. The following dilutions were used: anti-HNF4[11]: 1/50, anti-GFP (Cell Signaling Technology, RRID:AB_3101977): 1/500 (v/v). Tissues were rinsed twice in PBST and washed in PBS for 1 h using orbital shaking. They were incubated in secondary antibodies (Jackson ImmunoResearch Labs, RRID:AB_2340460 and RRID:AB_2315777) diluted 1/500 (v/v) in blocking buffer for 2 h in the dark at room temperature. Tissues were then rinsed twice in PBST and washed for 1 h in PBS before mounting in RotiMount FluorCare DAPI mounting medium (Carl Roth, Cat#HP20.1). For fluorescent staining of neutral lipids and filament

actin, tissues were permeabilized after fixation in PBST for 1 h, and incubated for 1 h at 22 °C in a solution containing BODIPY 493/503 (Invitrogen, Cat#D3922) and Alexa Fluor 555 Phalloidin (Cell Signaling Technologies, Cat#8953S), diluted 1/500 and 1/200 (v/v) in PBS, respectively. They were then washed thrice in PBS and mounted in RotiMount FluorCare DAPI mounting medium (Carl Roth, Cat#HP20.1). For Oil Red O stains, tissues were fixed as described above, washed twice in PBS and twice in propylene glycol (Acros Organics, Cat#ACRO447415000). Tissues were then incubated in a preheated (60 °C) solution of 0.5% Oil Red O (Sigma Aldrich, Cat#O0625-100G) dissolved in propylene glycol. Incubation was performed for 1 h at 37 °C without agitation. Following incubation, samples were washed twice in propylene glycol and twice in PBS before being mounted in glycerol. For all samples, tissues were imaged using either a BX53 brightfield and fluorescence microscope with the cell-Sens Standard Software (Olympus, v1.11) or a LSM 710 confocal microscope with the Zen 2009 software (Zeiss). Two to three independent stains were performed on sets of 5–15 tissues. A representative image is shown in the figures.

### Oil red O staining quantification

Pictures of Oil Red O-stained midguts were processed using Fiji (v.1.54p, RRID:SCR_002285). The ROI of the midgut region was drawn by hand, after which images were color deconvoluted using the H&E 2 vector. The Color_2 channel was converted to an 8-bit image and inverted. To detect consistent staining intensities, a threshold was determined for each set, depending on the illumination settings and was applied to every image to detect positively stained area. Finally, the positively stained area was divided by the area of the corresponding ROI to quantify the percentage of area stained with Oil red O.

### Analysis of nuclear size and integrated DAPI density

To measure DAPI integrated density, fixed midguts were mounted in confocal dishes (VWR, Cat#734-2906) in RotiMount FluorCare DAPI mounting medium (Carl Roth, HP20.1). TIFF images were acquired by confocal imaging over a 212.55 * 212.55 μm section with a 2 μm slice interval. Midgut regions were imaged from the upper surface down to the frontal plane. Using Fiji (v.1.54p, RRID:SCR_002285), individual nuclei were outlined by creating a Maximum Intensity Projection, detecting particles larger than 55 μm$^2$ and performing a binary transformation and watershed separation. An additional manual filtering step was performed to select complete and non-overlapping nuclei, and create nuclear regions of interest (ROI) to measure individual nucleus area. For integrated DAPI density, we created a Sum Slices Projection of the DAPI stack image and measured integrated density of individual nucleus. Nuclear areas and DAPI integrated density were averaged to obtain one representative value per midgut region.

### Measure of gut volume

Pictures from Oil Red O-stained midguts were also used to quantify gut volume. The Stitching plugin from Fiji (v.1.54p, RRID:SCR_002285) with the Maximal Intensity fusion method[70] was used for processing tiled images. For the measurement of gut volume, the InteredgeDistance macro (v2.0) (https://sourceforge.net/projects/interedgedistance/) was used to calculate the length and average diameter. Midgut volume was calculated using the formula for a cylinder $V = \pi * length * (\frac{diameter}{2})^2$.

### Graphical representation and statistical analysis

Graphical representation and statistical analysis were performed using GraphPad Prism software (v10, RRID:SCR_002798). We used heatmaps and min-max normalization to display transcript levels determined by RT-qPCR and mRNA-seq. Normalized transcript levels ($x_{norm}$) were determined for each sample using the following formula: $x_{norm} = (x-min)/(max-min)$ where $x$ is the actual transcript level determined by RT-qPCR or mRNA-seq, and min and max are the minimum and maximum values of all transcript levels across samples, respectively. For analyses comparing two or three groups, a Shapiro-Wilk normality test was performed to choose between parametric Student's $t$ test and 1-way ANOVA, or non-parametric Mann-Whitney and Kruskall-Wallis test. All other details about statistical analyses are included in the figure legends and in the source data file.

### Reporting summary

Further information on research design is available in the Nature Portfolio Reporting Summary linked to this article.

## Data availability

mRNA-seq data generated in this study have been deposited in the Gene Expression Omnibus (GEO) database under the following accession numbers: GSE283125 and GSE315710. Source data are provided with this paper.

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

## Acknowledgements

The authors thank FlyBase[67], the Vienna *Drosophila* Resource Center (VDRC; Vienna, Austria), the Bloomington *Drosophila* Stock Center (BDSC; Bloomington, IN, USA) supported by the NIH (NIH P40OD018537) and the Transgenic RNAi Project (TRiP). The authors also thank P. Leopold, F. Leulier and M. Uhlirova for sharing reagents; Ş. Delipınar, L. Helfer, Ö. Şeşen, the CECAD imaging facility, A. Abdallah and the CECAD bioinformatic facility and the Cologne Center for Genomics for technical support; F. Leulier for his comments on the manuscript. This work was funded by the Deutsche Forschungsgemeinschaft (DFG, German Research Foundation) under Germany's Excellence Strategy EXC 2030-390661388 (G.S.). This work was funded by the Deutsche Forschungsgemeinschaft (DFG, German Research Foundation), Project number 561529502 (G.S.).

## Author contributions

Conceptualization, Methodology, Visualization, Writing – original and revised draft: C.L. and G.S.; Funding acquisition, Project administration, Supervision: G.S.; Data curation, Formal analysis: A.F., C.L., and G.S., Investigation: C.L.

## Funding

## Competing interests

The authors declare no competing interests.
