## [Transparent Peer Review file · Nature Communications]

Transient remodeling of gut metabolism supports juvenile growth and adult fitness in *Drosophila*

Corresponding Author: Professor Gilles Storelli

Version 0:

Reviewer comments:

Reviewer #1

(Remarks to the Author)

This study by Lefranc and Storelli reveals that the *Drosophila* larval midgut undergoes a metabolic switch during early L3 development that supports a period of rapid growth. Using a transcriptomic approach, the authors demonstrate that genes associated with lipid metabolism, digestion, and trafficking are significantly elevated in early L3 larval midguts. Subsequent mechanistic studies demonstrate that this metabolic program is triggered by ecdysone signaling and mediated by Hepatocyte Nuclear Factor 4 (HNF4). Finally, the authors demonstrate that activation of this metabolic program is essential, as disruption of this signaling axis in enterocytes using either a dominant-negative EcR construct or HNF4-RNAi impaired growth, delayed maturation, and compromised adult traits such as desiccation resistance and fecundity.

Overall, I found the study to be straight forward and quite important. As noted both here and elsewhere, the rate of larval growth is significantly elevated during the final larval instar. This observation long hinted at the possibility of significant metabolic changes during this stage of development. Here, Lefranc and Storelli demonstrate that this is indeed the case and describe the molecular mechanism that activates the intestinal increase in lipid metabolism. In the future, this study will be seen as an important advance in the field of developmental metabolism – I imagine it will be highly cited.

I have only a few minor requests prior to publication:

- The introduction is a bit short and I'd ask that the authors provide additional background information. I'd suggest that priority be given to enhance background knowledge of HNF4.
- Similarly, I found the discussion to be lacking and the last three sections to be notably speculative and seemingly outside the scope of the study. While I think a discussion of juvenile hormone, gut bacteria, and Blimp-1 to be interesting, a more focused discussion would benefit the manuscript overall.
- In Figures 5, 6, and 7, notation for HNF4-RNAi is confusing, as the immediate impression upon examining the figure is that HNF4 is being overexpressed using *mex-Gal4*. Please make this point clearer.

I'd also suggest the following grammatical corrections:

Line 16- "but these regulations remain ill-defined." The use of the word regulation in this sentence is awkward – considered a word such as mechanisms instead.

Reviewer #2

(Remarks to the Author)

The story builds on the finding that *Drosophila* larval growth is not linear but highly accelerated at a transition phase between second and third instar. The authors find, through RNA sequencing experiments, that enhanced gut lipid metabolism correlate with the rapid systemic growth at this stage. In addition, they find the steroid hormone Ecdysone signaling as well as the *Drosophila* Hepatocyte Nuclear Factor 4 α (Hnf4) as mediators of the gut metabolic activity. Through genetic experiments, the authors show that inhibiting Ecdysone signaling or Hnf4 leads to the inability to mobilize gut lipids to circulation and reduced growth rate. Finally, the authors demonstrate a link between the enhanced lipid metabolism during larval growth and adult fitness traits.

The nutritional and metabolic requirements for *Drosophila* juvenile (larval) growth have been extensively studied. In addition, the role of steroid (ecdysone) signaling and Hnf4 as a regulator of lipid metabolism is known. However, the study of Lefranc & Storelli elegantly builds on the previous work to demonstrate an unprecedented physiological growth regulation at a specific point in larval development. Thus, this work significantly enhances our understanding of this important and physiologically relevant model of animal growth. Overall, I find this study to be important, and interesting to the readers of Nature Communications.

The manuscript is technically solid. It is very well written, and the data is presented with utmost care. The conclusions are, for the most part, well supported by the presented data. However, to make some of the key conclusions more convincing, I would ask the authors to add few more pieces of evidence to the manuscript. Please find my specific suggestions below.

Figures 4-6.

The authors use a dominant negative EcR and Hnf4 RNAi to inhibit ecdysone signaling and activation of Hnf4 target genes in the larval enterocytes, respectively. In both cases, it is shown that circulating lipid levels and systemic TAG stores are attenuated whereas the gut lipid levels are increased (Figures 4 and 5). This shows that EcR and Hnf4 are involved in mobilization of lipids to circulation at this point of larval development. In addition, it is also shown that larval growth rate is reduced (Figure 6). The authors also show that expression of the EcR-DN in the ECs led to reduced Hnf4 expression (Figure 5J). Based on this the authors suggest a model where Ecdysone signaling functions through Hnf4 in this setting. (In row 159 it is concluded "HNF4 acts downstream of EcR to induce intestinal lipid metabolism".) However, to show that Hnf4 indeed explains the phenotype of inhibiting ecdysone signaling, a genetic interaction experiment is warranted. The authors should simultaneously express EcR-DN and HNF4 in enterocytes and look for a rescue in the systemic lipid levels and growth rate. In addition, since the manuscript relies on the assumption that ecdysone functions through Hnf4, some evidence is needed to demonstrate that this interaction is direct. For example, are there regulatory elements in the Hnf4 promoter for mediators of Ecdysone signaling? If Hnf4 alone does not explain the phenotypes of EcR, the conclusions of the study should be diluted to allow a broader interpretation of the results.

Minor comment:

Figure 2.

The GO terms related to ribosome biogenesis are down at the point of rapid growth. This is somewhat surprising observation given that ribosome activity is essential for growth. How does this observation relate to the growth of the intestine itself at this point of larval development? Could the authors comment on this, for example, in the discussion?

Reviewer #3

(Remarks to the Author)

This manuscript explores how larval gut remodeling influences growth during development and fitness in adulthood, with a focus on the role of Hnf4-mediated ecdysone signaling. The experimental approaches used were logical and appropriate resulting in quality data, with a minor concern regarding clarity of sample size descriptions. Overall, the clearly presented data supported authors' interpretations and conclusions; however, there is a concern regarding showing a direct link between the EcR and Hnf4. The work described in this manuscript advances understanding of *Drosophila* developmental biology and adult physiology by describing the role that the gut plays. In addition, given the cellular and molecular similarities to the mammalian gut, this work may provide insight into how gut development in humans might impact fetal growth.

Below are specific concerns categorized as major, moderate, and minor.

Major

- Line 159: The conclusion is too strong. The data shows correlation (as stated in the text of this section). A genetic interaction experiment in which EcR activity is reduced while simultaneously overexpressing Hnf4 and seeing rescue of the phenotype would show more direct evidence that Hnf4 works downstream of EcR in this case.
- Lines 182-193, Figure 6: While the statistical analysis shows significance, the differences are very slight. Therefore, some discussion of how the differences are biologically relevant is needed.

Moderate

- Lines 70-79 and Figure 1: If I am interpreting what is written correctly, 11 biological replicates over two independent experiments means 5-ish larvae were assessed per experiment? This seems super low.
- Line 83-84 and Supp Fig 1: Is this assay best way to demonstrate that there is a difference in feeding amount between the larval stages? Perhaps this can be a more supportive piece than the rationale to rule out feeding.
- Lines 120-122 and Figure 3: The description of the sample size numbers and independent experiments is unclear.
- Line 149-150: In Figure 4E, the whole EcR-DN gut images shown appear to be larger than controls (at least from these two representative images), yet the text concludes that there is no difference in size.
- Line 188: It is not accurate to say "...fail to accumulate protein after entering the L3..." – they do accumulate from L2 to L3, but not as much as controls.

Minor

- Line 83: The first sentence needs to be referenced.
- Lines 93-101 and Figure 2A: It would be information to mention the types of genes found in cluster C (genes that don't

really change) and why that makes sense

- Line 118: The first mention of “different genetic backgrounds” was made here but was also assessed previously but not mentioned. To strengthen the conclusions made for this set of data, it might be better to emphasize the fact that the similar results are observed in different genetic backgrounds

- Figures 3F, 4E and 5E: Some way to point out where readers should focus (box, inset, outline, etc) would be nice. Maybe add some fluorescence images to show since line 125 says “intracellular” but these images aren’t at that level.

- Figure 7G: In the ovary images, anterior should be at the top. To obtain images in which gross ovarian morphology is more easily represented, perhaps ovaries can be imaged in the dissecting dish, submerged in media, instead of mounted on slides.

Reviewer #4

(Remarks to the Author)

Version 1:

Reviewer comments:

Reviewer #1

(Remarks to the Author)

The authors have addressed my concerns.

Reviewer #2

(Remarks to the Author)

The authors have addressed all of my comments/concerns.

Congratulations to the authors for their very nice study.

Reviewer #3

(Remarks to the Author)

I appreciate that the authors seriously considered all of my comments. They have put forth time and effort to address reviewer concerns and make appropriate revisions to strengthen the manuscript.

Revisions made to address the reviewers' comments

C. Lefranc, A. Fichant, G. Storelli

"Transient remodeling of gut metabolism supports juvenile growth and adult fitness in *Drosophila*"

NCOMMS-25-69350-T

We thank the reviewers for their supportive remarks and comments on our paper. Their suggestions were helpful and the resulting changes have, we feel, significantly improved the manuscript. Below we respond to each comment raised by the reviewers.

REVIEWER COMMENTS

REVIEWER #1 (REMARKS TO THE AUTHOR):

This study by Lefranc and Storelli reveals that the *Drosophila* larval midgut undergoes a metabolic switch during early L3 development that supports a period of rapid growth. Using a transcriptomic approach, the authors demonstrate that genes associated with lipid metabolism, digestion, and trafficking are significantly elevated in early L3 larval midguts. Subsequent mechanistic studies demonstrate that this metabolic program is triggered by ecdysone signaling and mediated by Hepatocyte Nuclear Factor 4 (HNF4). Finally, the authors demonstrate that activation of this metabolic program is essential, as disruption of this signaling axis in enterocytes using either a dominant-negative EcR construct or HNF4-RNAi impaired growth, delayed maturation, and compromised adult traits such as desiccation resistance and fecundity.

Overall, I found the study to be straight forward and quite important. As noted, both here and elsewhere, the rate of larval growth is significantly elevated during the final larval instar. This observation long hinted at the possibility of significant metabolic changes during this stage of development. Here, Lefranc and Storelli demonstrate that this is indeed the case and describe the molecular mechanism that activates the intestinal increase in lipid metabolism. In the future, this study will be seen as an important advance in the field of developmental metabolism – I imagine it will be highly cited.

We thank the reviewer for their support of our study.

I have only a few minor requests prior to publication:

- The introduction is a bit short and I'd ask that the authors provide additional background information. I'd suggest that priority be given to enhance background knowledge of HNF4.

As requested by the reviewer, we have added a paragraph to the introduction to provide additional background information on HNF4 (lines 45–57).

- Similarly, I found the discussion to be lacking and the last three sections to be notably speculative and seemingly outside the scope of the study. While I think a discussion of juvenile hormone, gut bacteria, and Blimp-1 to be interesting, a more focused discussion would benefit the manuscript overall.

In response to the reviewer's comments, we have made significant revisions to the Discussion.

In particular, we have:

- added more details on the direct and indirect ways in which gut metabolic remodeling could promote growth, and discussed the overall impact of this metabolic switch on systemic anabolism and adult fitness (lines 260-277)
- added a paragraph on the epistatic relationship between EcR, HNF4 and intestinal lipid metabolism (lines 279-293)

- added a paragraph on potential connections between ribosomal biogenesis and enterocyte function during development (lines 324-335). This was requested by reviewer #2, please see the comments below.

These changes result in an overall increase in the length of the discussion, while keeping it within the allotted space.

We would like to retain the sections on the timing of the metabolic switch (including the reference to juvenile hormone, lines 295-309) and the trade-offs between maximal gut metabolism and juvenile health (lines 311-322). These points are frequently raised by colleagues when we present these data, and they are scientific questions that we will actively pursue in the future.

We also wish to retain the section on the conservation of the pathways involved in intestinal metabolic remodeling to emphasize the broader significance of our findings.

Nonetheless, we have revised these sections to be more concise and reduce their relative importance in the Discussion.

- In Figures 5, 6, and 7, notation for HNF4-RNAi is confusing, as the immediate impression upon examining the figure is that HNF4 is being overexpressed using *mex-Gal4*. Please make this point clearer.

We have changed the labelling “*Hnf4i*” to “*Hnf4 RNAi*” to facilitate the interpretation of the figures.

I’d also suggest the following grammatical corrections:

Line 16- "but these regulations remain ill-defined." The use of the word regulation in this sentence is awkward – considered a word such as mechanisms instead.

We have rewritten this sentence to make it clearer (lines 15–16).

Reviewer #2 (Remarks to the Author):

The story builds on the finding that *Drosophila* larval growth is not linear but highly accelerated at a transition phase between second and third instar. The authors find, through RNA sequencing experiments, that enhanced gut lipid metabolism correlate with the rapid systemic growth at this stage. In addition, they find the steroid hormone Ecdysone signaling as well as the *Drosophila* Hepatocyte Nuclear Factor 4 α (Hnf4) as mediators of the gut metabolic activity. Through genetic experiments, the authors show that inhibiting Ecdysone signaling or Hnf4 leads to the inability to mobilize gut lipids to circulation and reduced growth rate. Finally, the authors demonstrate a link between the enhanced lipid metabolism during larval growth and adult fitness traits.

The nutritional and metabolic requirements for *Drosophila* juvenile (larval) growth have been extensively studied. In addition, the role of steroid (ecdysone) signaling and Hnf4 as a regulator of lipid metabolism is known. However, the study of Lefranc & Storelli elegantly builds on the previous work to demonstrate an unprecedented physiological growth regulation at a specific point in larval development. Thus, this work significantly enhances our understanding of this important and physiologically relevant model of animal growth. Overall, I find this study to be important, and interesting to the readers of Nature Communications.

The manuscript is technically solid. It is very well written, and the data is presented with utmost care. The conclusions are, for the most part, well supported by the presented data. However, to make some of the key conclusions more convincing, I would ask the authors to add few more pieces of evidence to the manuscript. Please find my specific suggestions below.

We thank the reviewer for their support of our study.

Figures 4-6.

The authors use a dominant negative EcR and Hnf4 RNAi to inhibit ecdysone signaling and activation of Hnf4 target genes in the larval enterocytes, respectively. In both cases, it is shown that circulating lipid levels and systemic TAG stores are attenuated whereas the gut lipid levels are increased (Figures 4 and 5). This shows that EcR and Hnf4 are involved in mobilization of lipids to circulation at this point of larval development. In addition, it is also shown that larval growth rate is reduced (Figure 6). The authors also show that expression of the EcR-DN in the ECs led to reduced Hnf4 expression (Figure 5J). Based on this the authors suggest a model where Ecdysone signaling functions through Hnf4 in this setting. (In row 159 it is concluded “HNF4 acts downstream of EcR to induce intestinal lipid metabolism”.) However, to show that Hnf4 indeed explains the phenotype of inhibiting ecdysone signaling, a genetic interaction experiment is warranted. The authors should simultaneously express EcR-DN and HNF4 in enterocytes and look for a rescue in the systemic lipid levels and growth rate.

To determine the extent to which HNF4 directs intestinal metabolic remodeling, we performed new bulk mRNA-seq in L2 and L3 midguts with *Hnf4* silencing in ECs. As previously observed, numerous genes involved in lipid metabolism are transcriptionally induced in control midguts upon entry into the L3 stage. 45% of these genes depend on HNF4 for their induction (New Fig. 5d, Supplementary Data 5, lines 176-180). In parallel, *Hnf4* overexpression in otherwise wild-type ECs further increases the expression of several of these lipid-related genes in L3 midguts (New Supplementary Fig. 5c, lines 180-183). Therefore, HNF4 drives a substantial part of intestinal metabolic remodeling during larval development.

We have then inhibited ecdysone signaling and simultaneously overexpressed *Hnf4* in ECs, and scored systemic lipid levels and biomass (using protein as an indicator of this parameter). HNF4 is necessary for fat storage in L3 larvae (Fig. 5h). However, *Hnf4* overexpression is not sufficient to rescue systemic fat accumulation when ecdysone signaling is inhibited in ECs (New Fig. 5j). Similar observations were made regarding biomass (New Supplementary Fig. 6c). Together with the fact that HNF4 induces parts of the developmental switch in lipid metabolism (New Fig. 5d, Supplementary Data 5, lines 176-180), these data support a model in which EcR acts through multiple pathways, including HNF4 signaling, to induce intestinal metabolic remodeling and accelerate fat storage and growth in L3 larvae. These new observations are discussed in details (lines 279-293).

In addition, since the manuscript relies on the assumption that ecdysone functions through Hnf4, some evidence is needed to demonstrate that this interaction is direct. For example, are there regulatory elements in the Hnf4 promoter for mediators of Ecdysone signaling?

Our data show that ecdysone signaling induces *Hnf4* expression in the midgut during larval development (Fig. 5i). These findings are in agreement with a previous study in mosquitoes, which demonstrates that steroid signaling can regulate lipid metabolism by promoting *Hnf4* expression (DOI: 10.1073/pnas.1619326114).

We analyzed publicly available ChIP-seq data to determine if ecdysone-related transcription factors bind in the vicinity of the *Hnf4* locus (New Supplementary Data 6). These analyses indicate that EcR and its partner USP bind several times in the vicinity of the *Hnf4* locus (New Supplementary Data 6). The same applies to Eip74EF, Eip75B, Eip93F, Blimp-1 and crol, which are well-established transcriptional targets of EcR. Therefore, EcR may act in direct and indirect manners to induce *Hnf4* expression during larval development. These new observations are detailed in (lines 194-198).

If Hnf4 alone does not explain the phenotypes of EcR, the conclusions of the study should be diluted to allow a broader interpretation of the results.

As explained above, our new mRNA-seq data show that HNF4 drives parts of the metabolic switch, controlling 45% of the lipid-related transcripts that are induced during development (New Fig. 5d, Supplementary Data 5, lines 176-180). This partial control may explain why *Hnf4* overexpression alone

does not rescue fat levels and systemic growth when ecdysone signaling is inhibited. We have revised parts of the results and discussion to incorporate these new data, and adjusted the conclusions of our study accordingly (please see response to previous comments).

Minor comment:

Figure 2.

The GO terms related to ribosome biogenesis are down at the point of rapid growth. This is somewhat surprising observation given that ribosome activity is essential for growth. How does this observation relate to the growth of the intestine itself at this point of larval development? Could the authors comment on this, for example, in the discussion?

We made similar observations with our new RNA-seq analysis on a different set of controls (*mex>atp40*, Supplementary Data 5). These effects are consistent with the inhibition of Myc, a master regulator of ribosome biogenesis, by ecdysone during larval development (DOI: 10.1016/j.devcel.2010.05.007). Despite the suppression of ribosome biogenesis, the midgut still grows between the L2 and L3 stage (Supplementary Fig. 3b, 3c, 3e and 3f). This paradox could be explained by the long half-life of these ribonucleoprotein particles, which usually exceeds several days (doi: 10.1146/annurev-cellbio-111822-113326). Therefore, ribosomes might be actively produced during L1 and L2 to cover for the needs of the last larval instar, which lasts approximately 48 hours under optimal conditions. The suppression of ribosome biogenesis in L3 could in turn facilitate the reallocation of resources to support EC-specific functions, such as nutrient digestion, absorption and export, thereby supporting systemic growth. These possibilities are now discussed in (lines 324 - 335).

REVIEWER #3 (REMARKS TO THE AUTHOR):

This manuscript explores how larval gut remodeling influences growth during development and fitness in adulthood, with a focus on the role of Hnf4-mediated ecdysone signaling. The experimental approaches used were logical and appropriate resulting in quality data, with a minor concern regarding clarity of sample size descriptions. Overall, the clearly presented data supported authors' interpretations and conclusions; however, there is a concern regarding showing a direct link between the EcR and Hnf4. The work described in this manuscript advances understanding of *Drosophila* developmental biology and adult physiology by describing the role that the gut plays. In addition, given the cellular and molecular similarities to the mammalian gut, this work may provide insight into how gut development in humans might impact foetal growth.

We thank the reviewer for their support of our study.

Below are specific concerns categorized as major, moderate, and minor.

Major

- Line 159: The conclusion is too strong. The data shows correlation (as stated in the text of this section). A genetic interaction experiment in which EcR activity is reduced while simultaneously overexpressing Hnf4 and seeing rescue of the phenotype would show more direct evidence that Hnf4 works downstream of EcR in this case.

Our data show that ecdysone signaling induces *Hnf4* expression in the midgut during larval development (Fig. 5i). These findings are in agreement with a previous study in mosquitoes, which demonstrates that steroid signaling can regulate lipid metabolism by promoting *Hnf4* expression (DOI: 10.1073/pnas.1619326114).

In parallel, analysis of publicly available ChIP-seq data indicates that EcR and its dimerization partner USP bind several times in the vicinity of the *Hnf4* locus (New Supplementary Data 6). The same applies to Eip74EF, Eip75B, Eip93F, Blimp-1 and *crol*, which are well-established transcriptional targets of

EcR. These data support that EcR acts in direct and indirect manners to induce *Hnf4* expression in the midgut during larval development. These new observations are detailed in (lines 194-200).

We also provide new RNA-seq analysis of midguts with *Hnf4 RNAi* with this revised manuscript (New Supplementary Data 5). As previously observed, numerous genes involved in lipid metabolism are transcriptionally induced in control midguts following the entry into the L3 stage. HNF4 is required for the induction of 45% of these genes (New Fig. 5d, Supplementary Data 5). In parallel, *Hnf4* overexpression in otherwise wild-type ECs further increases the expression of several lipid-related genes in L3 midguts (New Supplementary Fig. 5c). Therefore, HNF4 drives a substantial part of intestinal metabolic remodeling during larval development. These data are presented in a new section of the results, entitled “**HNF4 is necessary for intestinal metabolic remodeling**” (lines 169-188).

We have then inhibited ecdysone signaling and simultaneously overexpressed *Hnf4* in ECs, and scored systemic lipid levels and biomass (using protein as an indicator of the latter). HNF4 is necessary for fat storage in L3 larvae (Fig. 5h). However, *Hnf4* overexpression is not sufficient to rescue fat levels when ecdysone signaling is inhibited in ECs (New Fig. 5j). Together with the fact that HNF4 induces parts of the developmental switch in lipid metabolism (New Fig. 5d, Supplementary Data 5, lines 176-180), these data support a model in which EcR acts through multiple pathways, including HNF4 signaling, to drive intestinal metabolic remodeling and accelerate fat storage and systemic growth. These new observations are discussed in details (lines 279-293).

We agree that the former title of the section “**HNF4 acts downstream of EcR to induce intestinal lipid metabolism**” (line 159 of the former manuscript) could suggest that HNF4 is sufficient to rescue lipid metabolism under ecdysone inhibition. We therefore changed it to “**EcR acts via several pathways to support fat storage in L3**” for accuracy (line 190 of the revised manuscript).

- Lines 182-193, Figure 6: While the statistical analysis shows significance, the differences are very slight. Therefore, some discussion of how the differences are biologically relevant is needed.

The effects of EcR and HNF4 inhibition on body volume and systemic protein levels are less pronounced than those on triglyceride (Fig. 4h, 5h, 6a-d). Nevertheless, these differences are biologically relevant, as they result in smaller adults despite longer development (Figure 6e-f and 7b-d). It is also important to note that all of our experiments were conducted using a complete diet. Therefore, these differences may be further exacerbated when animals develop in suboptimal nutritional conditions, as would be encountered in the wild. Finally, additional physiological adaptations may act alongside gut metabolic remodeling to support increased anabolism during the L3 stage. We now discuss these points in (lines 271-277).

Moderate

- Lines 70-79 and Figure 1: If I am interpreting what is written correctly, 11 biological replicates over two independent experiments means 5-ish larvae were assessed per experiment? This seems super low.

In this case, a “biological replicate” consists of larvae collected from a vial that was initially seeded with 40 embryos. Therefore, 11 biological replicates would represent 440 larvae. We have updated the legend to clarify the nature of biological replicates. However, due to space constraints (figure legends are limited to 350 words), the exact contents of “biological replicates” (ie. number of tissues or animals) for each type of experiment is detailed in the Methods.

- Line 83-84 and Supp Fig 1: Is this assay best way to demonstrate that there is a difference in feeding amount between the larval stages? Perhaps this can be a more supportive piece than the rationale to rule out feeding.

We agree that the previous writing was confusing. We have rewritten this section for more clarity (line 88-91).

- Lines 120-122 and Figure 3: The description of the sample size numbers and independent experiments is unclear.

In these cases, a biological replicate contains several organs or hemolymph collected from multiple animals. We have updated the legend in Figure 3 to clarify the nature of the biological replicates (lines 959-960 and 965-966). The exact number of organs or animals used to make up one biological replicate is described in the methods, in the section “Sample processing for metabolite quantification” (lines 491-501) to comply with the space limitations for the figure legends (see comment above).

- Line 149-150: In Figure 4E, the whole *EcR-DN* gut images shown appear to be larger than controls (at least from these two representative images), yet the text concludes that there is no difference in size.

In the results, we state: “(*EcR-DN*) has global, negative impacts on the induction of transcripts involved in lipid metabolism in L3 midguts (Fig. 4d). Importantly, these effects are not due to decreased midgut growth or EC endoreplication (Supplementary Fig. 3b-f).”

We have measured gut volume, RNA and protein content in a large number of midguts, and found no difference between control and *EcR-DN* tissues (Supplementary Fig. 3b-d). We have also measured EC nuclear size and ploidy, and found a slight increase in nuclear size under *EcR-DN* expression, but only in the posterior midgut of L3 larvae (Supplementary Fig. 3e-f).

Therefore, the inhibition of gut metabolic remodeling in *EcR-DN* midguts is not explained by a general defect in midgut growth or maturation, as stated in the results.

We have used alternative images in (Fig. 4e) to align them more closely with the measurements shown in (Supplementary Fig. 3b-f).

- Line 188: It is not accurate to say “...fail to accumulate protein after entering the L3...” – they do accumulate from L2 to L3, but not as much as controls.

We have corrected this sentence (lines 215-216 in the revised version of the manuscript).

Minor

- Line 83: The first sentence needs to be referenced.

We have added a reference to one of James H. Sang’s seminal papers on *Drosophila* nutrition to support this claim (DOI:10.1242/jeb.33.1.45, line 88 in the revised version of this manuscript).

- Lines 93-101 and Figure 2A: It would be information to mention the types of genes found in cluster C (genes that don’t really change) and why that makes sense

Our gene ontology term enrichment analysis of Cluster C did not reveal any significant results – these genes do not appear to be involved in a common biological process (Fig. 2a and Supplementary Data 2, lines 104-105). Therefore, we are unable to comment on why the transcriptional suppression of this diverse set of genes is relevant to organismal development.

- Line 118: The first mention of “different genetic backgrounds” was made here but was also assessed previously but not mentioned. To strengthen the conclusions made for this set of data, it might be better to emphasize the fact that the similar results are observed in different genetic backgrounds

We have rewritten these sentences accordingly (lines 116 and 126).

- Figures 3F, 4E and 5E: Some way to point out where readers should focus (box, inset, outline, etc) would be nice. Maybe add some fluorescence images to show since line 125 says “intracellular” but these images aren’t at that level.

We have added arrowheads to those pictures, as well as to (Supplementary Fig. 3a and 5a) to highlight the regions of interest.

We also used confocal imaging to confirm the intracellular localization of the lipid droplets. A representative image can be seen in (New Supplementary Fig. 2b).

- Figure 7G: In the ovary images, anterior should be at the top. To obtain images in which gross ovarian morphology is more easily represented, perhaps ovaries can be imaged in the dissecting dish, submerged in media, instead of mounted on slides.

We have generated new images using the methods suggested by the reviewer and corrected their orientation. They are now shown in (Fig. 7g).

REVIEWER #4 (REMARKS TO THE AUTHOR):

We would like to thank the reviewer for supporting our study.

Please see our responses to the points raised by the reviewers 1–3.

Universität Heidelberg, COS Heidelberg, Im Neuenheimer Feld 230, 69120 Heidelberg

Heidelberg, 07.03.2026

Prof. Dr. Gilles Storelli
Department Leader
Physiology and symbiosis
Tel. +49 6221 54-6269
Gilles.storelli@cos.uni-heidelberg.de

Subject: Point-by-point response to the reviewers' comments

C. Lefranc, A. Fichant, G. Storelli. "Transient remodeling of gut metabolism supports juvenile growth and adult fitness in *Drosophila*". NCOMMS-25-69350A.

REVIEWER COMMENTS

Reviewer #1 (Remarks to the Author):

The authors have addressed my concerns.

Reviewer #2 (Remarks to the Author):

The authors have addressed all of my comments/concerns.
Congratulations to the authors for their very nice study.

Reviewer #3 (Remarks to the Author):

I appreciate that the authors seriously considered all of my comments. They have put forth time and effort to address reviewer concerns and make appropriate revisions to strengthen the manuscript.

We would like to thank the reviewers for taking the time to review our manuscript and for their supportive comments. Their suggestions were extremely helpful, and the resulting changes have greatly improved the manuscript.

Sincerely,

Gilles Storelli, on the behalf of all authors.